



# The Half-order Energy Balance Equation, Part 2: The inhomogeneous HEBE and 2D energy balance models

Shaun Lovejoy

5   Physics dept., McGill University, Montreal, Que. H3A 2T8, Canada

Correspondence: Shaun Lovejoy (lovejoy@physics.mcgill.ca)





**Abstract:** In part I, we considered the zero-dimensional heat equation showing quite generally that conductive – radiative surface boundary conditions lead to half-ordered derivative relationships between surface heat fluxes and temperatures: the Half-ordered Energy balance Equation (HEBE). The real Earth – even when averaged in time over the weather scales (up to ≈ 10 days) – is highly heterogeneous, in this part II, we thus extend our treatment to the horizontal direction. We first consider a homogeneous Earth but with spatially varying forcing. Using Laplace and Fourier techniques, we derive the Generalized HEBE (the GHEBE) based on half-ordered space-time operators. We analytically solve the homogeneous GHEBE, and show how these operators can be given precise interpretations.

We then consider the full inhomogeneous problem with horizontally varying diffusivities, thermal capacities, climate sensitivities and forcings. For this we use Babenko's operator method which generalizes Laplace and Fourier methods. By expanding the inhomogeneous space-time operator at both high and low frequencies, we derive 2-D energy balance equations that can be used for macroweather forecasting, climate projections and for studying the approach to new (thermodynamic equilibrium) climate states when the forcings are all increased and held constant.

## 1 Introduction

In part I, we showed that when the surface of a body exchanges heat both conductively and radiatively, that its flux depends on the half order derivative of the surface temperature. This implies that energy stored in the subsurface effectively has a huge power law memory. This contrasts with the usual phenomenological assumption used notably in box models (including zero dimensional global energy balance models) that the order of derivative is an integer (one) and that on the contrary, the memory is only exponential (short). The result followed directly by assuming that the continuum mechanics heat equation was obeyed and the depth of the media was of the order of a few diffusion depths, for the Earth, perhaps several hundred meters. The basic result was a classical application of the heat equation barely going beyond results that [*Brunt*, 1932] already found "in any textbook".

A consequence was that although Newton's law of cooling is obeyed, that the temperature obeyed the half-order energy balance equation (HEBE) rather than the phenomenological first order Energy balance Equation (EBE). When applied to the Earth, the HEBE and its implied long memory explains the success of both climate projections through to 2100 [*Hebert*, 2017], [*Lovejoy et al.*, 2017] and macroweather (monthly, seasonal) temperature forecasts [*Lovejoy et al.*, 2015], [*Del Rio Amador and Lovejoy*, 2019]. We also considered the responses to periodic forcings showing that surface heat fluxes and temperatures are related by a complex thermal impedance ($Z(\omega)$, $\omega$ is the frequency). In the Earth system, $Z(\omega) = \lambda(\omega)$ where $\lambda(\omega)$ is the complex climate sensitivity that we estimated from a simple semi-empirical model.

Although in part 1 we discussed the classical 1-D application of the heat equation to the Earth's latitudinal energy balance (Budyko-Sellers models) - especially their ad hoc treatment of the surface boundary condition – we restricted the discussion to zero horizontal dimensions. In this part II, we first (section 2) extend the part I treatment to horizontally homogeneous systems but with inhomogeneous forcings, we then consider the more realistic case of horizontally inhomogeneous media.





The homogeneous case is quite classical and can be treated with standard Laplace and Fourier techniques, it leads to the
(horizontally) Generalized HEBE: the GHEBE.  Although the GHEBE has a more complex fractional derivative operator, like
the HEBE, it can nevertheless be given precise meaning via its Green's function.

In section 3, we derive the inhomogeneous GHEBE and HEBE.  This done by using of Babenko's method [*Babenko*,
1986] which is essentially a generalization of the Laplace and Fourier transform techniques.  The challenge with Babenko's
method is to interpret the inhomogenous fractional operators.  Following Babenko, we do this using both high and low
frequency expansions corresponding respectively to processes dominated by storage and by horizontal heat transport.  The
long time limit describes the new climate state that results when the forcing is increased everywhere and held fixed.   We also
include several appendices focused on empirical parameter estimates (appendix A), the implications for two point and space-
time temperature statistics (when the system is stochastically forced, internal variability, appendices B, C), and finally
(appendix D), the changes needed to account for the Earth's spherical geometry, including the definition of fractional operators
on the sphere.

## 2. The two-dimensional homogeneous heat equation

### 2.1 The homogeneous GHEBE

In part I we recalled the heat equation for the time-varying temperature anomalies ($T$) with diffusive and (horizontal) effective
advective velocity ($\underline{v}$):

$$\left(\frac{\partial}{\partial t}-\kappa_v\frac{\partial^2}{\partial z^2}\right)T=-\underline{v}\cdot\nabla_h T+\kappa_h\nabla_h^2 T$$

(1)

(This is written in the still general form of eq. 19, part I).  $\kappa_h$, $\kappa_v$ are horizontal and vertical thermal diffusivities, $z$ the vertical
coordinate (pointing upwards, the Earth is $z\leq 0$), $t$ the time, $\underline{x}=(x,y)$ the horizontal coordinates, $\nabla_h=\hat{x}\partial/\partial x+\hat{y}\partial/\partial y$ (the
circonflexes indicate unit vectors).  These equations must now be solved using the conductive-radiative surface boundary
condition:

$$\left(\frac{T(\underline{x},z,t)}{\lambda}+\rho c\kappa_v\frac{\partial T(\underline{x},z,t)}{\partial z}\right)\Bigg|_{z=0}=F(\underline{x},t)$$

(2)

$\rho$, $c$ are the fluid densities and specific heats $\lambda$ is the climate sensitivity and $F$ is the anomaly forcing.  The initial conditions
are $T=0$ at $z=-\infty$ (all $t$), and $T(\underline{x},z,t=0)=0$ (Riemann-Liouville) or below, $T(\underline{x},z,t=-\infty)=0$ (Weyl).



In part I, we nondimensionalized the zero-dimensional homogeneous operators by nondimensionalizing time by the relaxation

time: $t \rightarrow t/\tau$ (with $\tau = \kappa_v (\rho c \lambda)^2$) and nondimensionalizing the vertical distance by the vertical diffusion depth:

$z \rightarrow z/l_v$, with $l_v = (\tau \kappa_v)^{1/2}$. Considering now the full equation with advective and diffusive transport, we

nondimensionalize the horizontal coordinates by the horizontal diffusion length: $\underline{x} \rightarrow \underline{x}/l_h$, (with $l_h = (\tau \kappa_h)^{1/2}$) and use

the nondimensional advection velocity $\underline{\alpha} = \dfrac{\underline{v}}{V}$ (with speed $V = \dfrac{l_h}{\tau}$). If we now take $\lambda = 1$ (equivalent to using dimensions

of temperature for the forcing $F$), we obtain:

$$\left( \frac{\partial^2}{\partial z^2} - \left( \frac{\partial}{\partial t} + \left(-\nabla_h^2\right) - \underline{\alpha} \cdot \nabla_h \right) \right) T = 0$$

$$\left. \frac{\partial T}{\partial z} \right|_{z=0} + T(t,\underline{x};0) = F(t,\underline{x}) \tag{3}$$

For the heat equation and the conductive-radiative surface boundary condition respectively. For initial conditions such that $T$ = 0 for $t \leq 0$, as in part I, we take Laplace transforms in time, but we now take Fourier transforms in the horizontal:

$$\left( \frac{\partial^2}{\partial z^2} - \left( \frac{\partial}{\partial t} + \left(-\nabla_h^2\right) - \underline{\alpha} \cdot \nabla_h \right) \right) T = 0 \quad \overset{L.T.(t),F.T.(\underline{x})}{\longleftrightarrow} \quad \left( \frac{d^2}{dz^2} - \left( p + k^2 - i\underline{\alpha} \cdot \underline{k} \right) \right) \hat{T} = 0 \tag{4}$$

Where "F.T." is the Fourier transform in horizontal space, $\underline{k}$ for the conjugate of $\underline{x}$, $k = |\underline{k}|$ (the vector modulus) with

conjugate variable $r = |\underline{x}|$ (as usual, $\nabla_h \overset{F.T.}{\longleftrightarrow} i\underline{k}$). Fourier transforms in space are convenient for either infinite horizontal

media, or media with periodic horizontal boundary conditions. In appendix D, we consider the changes needed to account for

spherical geometry.

When $F(t,\underline{x}) = \delta(t)\delta(\underline{x})$, the solution $T(t,\underline{x}) \rightarrow G_\delta(t,\underline{x})$ and $\hat{T}(p,\underline{k}) \rightarrow \hat{G}_\delta(p,\underline{k})$ where $G_\delta$ is the impulse

(Dirac) response Green's function, part I, eq. 30. From eq. 4, we see that this is the same as the zero dimensional equation

(eq. 24, part I) but with $p \rightarrow p + k^2 - i\underline{\alpha} \cdot \underline{k}$ i.e. for the corresponding Green's function:

$$\widehat{G_\delta}(p,k;z) = \widehat{G_\delta}\left( p + k^2 - i\underline{\alpha} \cdot \underline{k};z \right) \tag{5}$$





A note on notation: the first argument is time, with the vertical separated by a semi-colon. When there is a horizontal coordinate

it comes after time, before the semicolon. With this notation, the right hand side of eq. 5 is the L.T. of the zero-dimensional

(time-depth) Green's function $G_\delta(t;z)$, the left hand side is the Laplace (time) and Fourier transform (horizontal, space)

transform.

We can now use the basic Laplace shift property:

$$e^{\left(-k^2+i\underline{\alpha}\cdot\underline{k}\right)t}G_\delta(t;z) \overset{L.T.(t)}{\leftrightarrow} \widehat{G_\delta}\left(p+k^2-i\underline{\alpha}\cdot\underline{k};z\right) \tag{6}$$

To conclude that:

$$\hat{G}_\delta(t,\underline{k};z) = e^{\left(-k^2+i\underline{\alpha}\cdot\underline{k}\right)t}G_\delta(t;z) \tag{7}$$

Decomposing this into a circularly symmetric diffusion part $\widehat{G}_{\delta,dif}(t,k;z)$ and a factor $e^{i\underline{k}\cdot\underline{\alpha}t}$ that shifts phases, we obtain:

$$\hat{G}_\delta(t,\underline{k};z) = e^{i\underline{k}\cdot\underline{\alpha}t}\widehat{G}_{\delta,dif}(t,k;z); \quad \widehat{G}_{\delta,dif}(t,k;z) = e^{-k^2t}G_\delta(t;z) \tag{8}$$

By circular symmetry of $\widehat{G}_{\delta,dif}(t,k;z)$, its inverse (2-D) Fourier transform reduces to an inverse Hankel transform ("H.T.").

Using:

$$\frac{e^{-r^2/(4t)}}{2t} \overset{H.T.}{\leftrightarrow} e^{-k^2t} \tag{9}$$

We therefore obtain for the diffusive part of the surface impulse response (i.e. the response with source spatial forcing

$\delta(\underline{x}) = \delta(r)/(2\pi r)$ ):

$$G_{\delta,dif}(t,r;z) = \frac{e^{-r^2/(4t)}}{2t}G_\delta(t;z) \tag{10}$$

Where $G_\delta(t;z)$ is the zero-dimensional impulse response. If needed, its integral representation is given in eq. 30, part I. The

last step is to take into account the advective term associated with the phase shift $\underline{k}\cdot\underline{\alpha}t$ . For this final step, we use the Fourier

shift theorem to obtain:





$$G_\delta(t,\underline{x};z) = G_{\delta,dif}(t,|\underline{x}-\underline{\alpha}t|;z) = \frac{e^{-|\underline{x}-\underline{\alpha}t|^2/(4t)}}{2t} G_\delta(t;z)$$

(11)

This is the general surface result for the diffusive-advective transport part of the spatially homogeneous case. As
expected, the advective transport simply displaces the center of the impulse response with nondimensional velocity $\underline{\alpha}$. As
usual, the solutions for arbitrary forcing $F(t,\underline{x})$ can be obtained by convolution.

For the surface we obtain the simpler expressions:

$$G_{\delta,dif}(t,r;0) = \frac{e^{-r^2/(4t)}}{2t}\left(\frac{1}{\sqrt{\pi t}} - e^t erfc\sqrt{t}\right)$$

$$G_{\Theta,dif}(t,r;0) = \int_0^t G_{\delta,dif}(t,r;0)dt = \frac{1}{r} erfc\left(\frac{r}{2\sqrt{t}}\right) - \int_0^t \frac{e^{-\frac{r^2}{4s}+s}}{2s} erfc(s^{1/2})ds$$

(12)

(see eq. 31, part I).  From these, the general surface results including advection are obtained with $r \to |\underline{x}-\underline{\alpha}t|$ , i.e.

$$G_\delta(t,\underline{x};0) = G_{\delta,dif}(t,|\underline{x}-\underline{\alpha}t|;0).$$

Since the advection term has this simple consequence, below we take $\underline{\alpha} = 0$, considering only diffusive transport, advection

can easily be included if needed (i.e. below, we take $G_\delta(t,r;0) = G_{\delta,dif}(t,r;0)$).

To better understand the impulse response, fig. 1 shows this surface $G_\delta(t,r;0)$ for various radial distances $r$ and fig. 2 shows

the corresponding time dependence of the time integral of $G_\delta$; the unit step response $G_\Theta$ for various distances $r$, illustrating the

power law approach to thermodynamic equilibrium at large $t$ (discussed in section 2.2).  The corresponding long time, short

distance expansions are:

$$G_\delta(t,r;0) \approx \frac{t^{-5/2}}{4\sqrt{\pi}} - \frac{(6+r^2)}{16\sqrt{\pi}}t^{-7/2} + O(t^{-9/2})$$

$$G_\Theta(t,r;0) \approx G_{therm,\delta}(r;0) - \frac{t^{-3/2}}{6\sqrt{\pi}} + \frac{(6+r^2)}{40\sqrt{\pi}}t^{-5/2} + O(t^{-7/2})$$

$\quad ; \qquad \begin{array}{l} t \gg 1 \\ r \ll 1 \end{array}$

(13)



Where $G_{therm,\delta}(r,0)$ is the Green's function for the (spatial Dirac) "hotspot" thermodynamic equilibrium response

discussed below (eq. 20). Note that the leading term in $G_{\delta}(t,r;0)$ is independent of $r$, and the leading term in the approach

to thermodynamic equilibrium $G_{\Theta}(t,r;0)$ is also independent of $r$.

Just as we derived the zero-dimensional HEBE by showing that it had the same Green's function as the $z = 0$ transport equation

Green's function, we can likewise derive the homogeneous Generalized Half-Order Energy Balance Equation (GHEBE) which

is the space-time surface equation whose Green's function is given in eq. 12. Following the derivation of the HEBE, in part I

eq. 29, and replacing $p \rightarrow p + k^2 - i\underline{\alpha} \cdot \underline{k}$ we obtain:

$$\hat{G}_{\delta}(p,\underline{k};z) = \frac{e^{\sqrt{p+k^2-i\underline{\alpha}\cdot\underline{k}}z}}{\sqrt{p+k^2-i\underline{\alpha}\cdot\underline{k}}+1} \tag{14}$$

Hence, for $z = 0$:

$$\left[\left(\frac{\partial}{\partial t}+\left(-\nabla_h^2\right)-i\underline{\alpha}\cdot\nabla_h\right)^{1/2}+1\right]G_{\delta}(t,\underline{x};0)=\delta(t)\delta(\underline{x}) \overset{(L.T.(t),F.T.(\underline{x}))}{\longleftrightarrow} \left(\sqrt{p+k^2-i\underline{\alpha}\cdot\underline{k}}+1\right)\hat{G}_{\delta}(p,\underline{k};0)=1 \tag{15}$$

The left hand equation is the homogeneous GHEBE whose Green's function is given by eq. 12. We have therefore found a

surprisingly simple explicit formula for the (inverse) half-order space-time GHEBE operator:

$$\left[\left(\frac{\partial}{\partial t}+\left(-\nabla_h^2\right)-i\underline{\alpha}\cdot\nabla_h\right)^{1/2}+1\right]^{-1}=G_{\delta}(t,\underline{x};0)* \tag{16}$$

where "$*$" indicates convolution. This allows us to give a precise interpretation of the half-order operator. Therefore the

dimensional, homogeneous, GHEBE and its full solution are:

$$\left(\tau\frac{\partial}{\partial t}+\left(-l_h^2\nabla_h^2\right)-i\underline{\alpha}\cdot\nabla_h\right)^{1/2}T_s(t,\underline{x})+T_s(t,\underline{x})=\lambda F(t,\underline{x})$$

$$T_s(t,\underline{x})=\lambda\int_{surf}\int_0^t G_{\delta}\left(\frac{t-t'}{\tau},\frac{|\underline{x}-\underline{x}'|}{l_h};0\right)F(t',\underline{x}')\frac{dt'}{\tau}\frac{d\underline{x}'}{l_h^2} \tag{17}$$

$$=\frac{\lambda}{l_h^2}\int_{surf}\int_0^t\frac{e^{-\tau|\underline{x}-\underline{x}'-\underline{\alpha}(t-t')|\tau|^2/(4l_h^2(t-t'))}}{2(t-t')}\left(\sqrt{\frac{\tau}{\pi(t-t')}}-e^{(t-t')/\tau}erfc\sqrt{\frac{(t-t')}{\tau}}\right)F(t',\underline{x}')dt'd\underline{x}'$$



("surf" is the surface over which the forcing acts, the bottom line uses the explicit eq. 12 for $G_\delta$).

The above shows that even with the purely classical integer-ordered Budyko-Sellers type heat equation, that surface temperatures already obey long memory, half order equations. However, it is not certain that the classical heat equation is in fact the most appropriate model. Straightforward generalizations to fractional heat equations - where $\tau \frac{\partial T}{\partial t} \rightarrow \tau^{2H} {}_\infty D_t^{2H} T$

lead directly to fractional energy balance equations for surface temperatures, we investigate fractional heat equations elsewhere. Physically, this generalization from the classical fractional value $H = 1/2$ could be a consequence of turbulent diffusive transport which since at least Richardson been known to have anomalous diffusion.

## 2.2 Thermodynamic equilibrium

If $F(t) = 0$, then the system is at equilibrium and will stay there. However, if $F$ is a step function in time, then as $t \rightarrow \infty$, a new equilibrium will be established. At equilibrium, $d/dt = 0$, so that the conjugate variable $p = 0$. With this and $\underline{\alpha} = 0$ in eq. 15, we obtain the equation for the (spatial) surface impulse response $G_{therm,\delta}(r;0)$ for thermodynamic equilibrium (subscript "therm"):

$$\left( \left( -\nabla_h^2 \right)^{1/2} + 1 \right) G_{therm,\delta} = \delta\left(\underline{x}\right) \overset{F.T.}{\leftrightarrow} \left( k + 1 \right) \hat{G}_{thermo,\delta} = 1 \tag{18}$$

i.e. the same as eq. 4 but with $p = 0$ (and $\underline{\alpha} = 0$) hence:

$$\hat{G}_{thermo,\delta}\left(k;z\right) = \frac{e^{kz}}{1+k} \tag{19}$$

The equilibrium surface temperature (spatial) impulse (Dirac "hotspot") Green's function is therefore:

$$G_{therm,\delta}\left(r,0\right) = \frac{1}{r} + \frac{\pi}{2}\left(Y_0\left(r\right) - H_0\left(r\right)\right) \overset{(H.T.)}{\leftrightarrow} \hat{G}_{thermo,\delta}\left(k;0\right) = \frac{1}{1+k} \tag{20}$$

Where $H_0$ is the zeroth order Struve function and $Y_0$ is the zeroth order Bessel function of the second kind. For large $r$, we

have the expansions:

$$G_{therm,\delta}\left(r;0\right) \approx \frac{1}{r^3} - \frac{9}{r^5} + O\left(r^{-7}\right); \quad r \gg 0 \tag{21}$$



$$G_{therm,\delta}(r;0) \approx \frac{1}{r} + \log r + \gamma_E - \log 2 - r + \frac{r^2}{4}(1 + \log 2 - \gamma_E) - \frac{r^2}{4}\log r + ...;$$

$$r \approx 0$$

The $1/r^3$ asymptotic decay is fast and implies that  spatial hotspots remain fairly localized; indeed, it is easy to show that if

instead we had a Dirac surface heat flux source driving the system (i.e. with surface BC $\left.\frac{\partial T}{\partial z}\right|_{z=0} = \delta(\underline{x})$ i.e. without radiation)

that the decay would be the much faster ($1/r$).  Forcing inhomogeneities thus remain much more localized than would otherwise
be the case.

 To study the convergence to thermodynamic equilibrium, consider a simple model of a surface "hot spot" where the forcing
is confined to a unit circle and turned on and held at a constant unit temperature at $t = 0$.  This is the spatial equivalent of a step

forcing in space, we combine it with a step (Heaviside) in time:

$$F(t,r) = \Theta(t)\Pi_1(r); \quad \Pi_1(r) = \begin{matrix} 1 & r \leq 1 \\ 0 & r > 1 \end{matrix} \tag{22}$$

$\Pi_1(r)$ is the corresponding indicator function. We now use the transform pair $\Pi_1(r) \overset{H.T.}{\leftrightarrow} \frac{J_1(k)}{k}$ to perform the convolution:

$$T_s(t,r) = G_\Theta(t,r;0) * \Theta(t)\Pi_1(r) \overset{H.T.}{\leftrightarrow} \frac{J_1(k)}{k}\hat{G}_\Theta(t,k;0) \tag{23}$$

($J_1$ is the first order Bessel function of first kind).  Taking the limit $t \rightarrow \infty$ we obtain the thermodynamic equilibrium

temperature distribution.  Alternatively we could find it directly by from eq. 19:

$$T_{therm,s}(r) = T_s(\infty,r) \overset{H.T.}{\leftrightarrow} \frac{J_1(k)}{k(1+k)} \tag{24}$$

Fig. 4 shows the cross section as a function of the distance from the circle's center at various times (the inverse Hankel
transforms were done numerically).  We note that the temperature rises very quickly at first, then slowly reaches equilibrium
(thick). The figure also shows (dashed) the thermodynamic equilibrium when the forcing is purely due to unit conductive

heating over the unit circle. The difference between the dashed  and the thick thermodynamic equilibrium curves are purely
due to the radiative loses in the latter.   (Note that in the zero-dimensional case (part I), using pure heating forcing boundary
conditions leads to diverging temperatures, there is no thermodynamic equilibrium.  This explains why Brunt instead used
temperature forcing boundary conditions.  Here, in two horizontal dimensions, boundary conditions that impose a fixed





temperature over the circle are problematic since they imply infinite horizontal temperature gradients and infinite horizontal

heat fluxes).

Figs. 5, 6 shows the same evolution but with temperature as a function of time for various distances (fig. 5) and as contours in space-time (fig. 6). We see that equilibrium is largely established in the first two relaxation times (here $\tau = 1$) and most of the perturbation is confined to two horizontal diffusion distances (here, $l_h = 1$).

## 3. The inhomogeneous heat equation

### 3.1 Babenko's method

The homogeneous heat equation in a semi-infinite domain is a classical problem and conductive- radiative surface boundary conditions naturally lead to fractional order operators, the HEBE and GHEBE. Although we have seen that fractional operators naturally, their advantages are much more compelling for the more realistic inhomogeneous equations relevant for the Earth. We will therefore now proceed to derive the inhomogeneous HEBE and GHEBE using Babenko's method. The more usual

application is to find the surface heat flux given a solution to the conduction equation (see for example [*Magin et al.*, 2004], [*Chenkuan and Clarkson*, 2018]), the following appears to be original.

In the inhomogeneous case with $\tau = \tau(\underline{x})$, $l_h = l_h(\underline{x})$, $l_v = l_v(\underline{x})$, $\underline{\alpha} = \underline{\alpha}(\underline{x})$, there is no unique nondimensionalization. Therefore, we express the inhomogeneous anomaly heat equation with nondimensional operators as:

$$\left(\tau\frac{\partial}{\partial t}+l_h\zeta-\left(l_v\frac{\partial}{\partial z}\right)^2\right)T=0; \qquad \zeta=\left(\underline{\alpha}\cdot\nabla_h+l_h\left(-\nabla_h^2\right)\right) \tag{25}$$

Where we have used $\kappa_v(\underline{x})=l_v^2\frac{\partial^2}{\partial z^2}=\left(l_v\frac{\partial}{\partial z}\right)^2$ and $\zeta$ is a time independent horizontal transport operator allowing for

both advective and diffusive transport. Under the fairly general conditions, when $\zeta$ operates on the temperature field, it is proportional to the nondimensional divergence of the horizontal heat flux (discussed in part I, see eq. 4). Since the forcing is via the surface boundary condition rather than by an inhomogeneous term, eq. 25 is mathematically homogeneous.

The first step in Babenko's method (see e.g. [*Podlubny*, 1999], [*Magin et al.*, 2004]), is to factor the differential operator:

$$\left(\Lambda+l_v\frac{\partial}{\partial z}\right)\left(\Lambda-l_v\frac{\partial}{\partial z}\right)T=0; \quad \Lambda=\left(\tau\frac{\partial}{\partial t}+l_h\zeta\right)^{1/2} \tag{26}$$

As usual, the general solution of a homogeneous equation is a linear combination of elementary solutions $A_+$ and $A_-$:





$$\left(\Lambda+l_v\frac{\partial}{\partial z}\right)A_+\left(t,\underline{x};z\right)=0; \quad \left(\Lambda-l_v\frac{\partial}{\partial z}\right)A_-\left(t,\underline{x};z\right)=0 \tag{27}$$

The $A_+$ solution leads to solutions that diverge at $z=-\infty$ whereas $A_-$ leads to the required physical solutions with $T\left(-\infty\right)=0$, ([*Podlubny*, 1999]). Therefore we are interested in solutions to:

$$\left(\Lambda-l_v\frac{\partial}{\partial z}\right)T\left(t,\underline{x};z\right)=0 \tag{28}$$

putting $z=0$ and using eq. 26, we obtain:

$$\left(\tau\frac{\partial}{\partial t}+l_h\zeta\right)^{1/2}T_s=l_v\frac{\partial T}{\partial z}\bigg|_{z=0}=\lambda Q_s; \quad \begin{array}{l} T_s\left(t,\underline{x}\right)=T\left(t,\underline{x};0\right) \\ Q_s\left(t,\underline{x}\right)=-\left(\underline{Q}_d\left(t,\underline{x};0\right)\right)_z \end{array} \tag{29}$$

where $T_s(t,\underline{x})$ is the surface temperature anomaly and $Q_s$ is the heat flux into the surface (the negative of $Q_{s,d,z}$ which is the $z$ component of the surface conductive (sensible) heat flux). Before interpreting the half order operator on the left, we can

already give this equation a physical interpretation. When $Q_s>0$, sensible heat is forced into the Earth, some of it is stored in

the subsurface (the $\tau\dfrac{\partial}{\partial t}$ term, the same horizontal position $\underline{x}$ but stored by heating up the subsurface, $z<0$), and some of the

heat (the $l_h\zeta$ term), is transported horizontally to neighbouring regions (and conversely when $Q_s<0$). We can also understand the basic difference between the $A_+$ and $A_-$ solutions: whereas the physically relevant $A_-$ solution correspond to energy storage and horizontal transport in the region $z<0$, the $A_+$ solutions correspond to the region $z>0$ assumed to be devoid

of conducting material.

The final step is to use the fact that the conductive heat flux $Q_s$ is equal to the radiative imbalance (part I, fig. 1):

$$Q_s=R_\uparrow-R_\downarrow=\frac{T_s}{\lambda}-F \tag{30}$$

Combining the equations 29, 30 we obtain the inhomogeneous Generalized Half-order Energy Balance Equation (GHEBE):

$$\left(\tau\left(\underline{x}\right)\frac{\partial}{\partial t}+l_h\left(\underline{x}\right)\zeta\left(\underline{x}\right)\right)^{1/2}T_s\left(t,\underline{x}\right)+T_s\left(t,\underline{x}\right)=\lambda\left(\underline{x}\right)F\left(t,\underline{x}\right) \tag{31}$$



If needed, the internal field $T(t,\underline{x};z)$, can be found by solving eq. 31 for $T_s(t,\underline{x})$ which is the $z = 0$ boundary condition for the full eq. 25. We see that eq. 31 reduces to the homogeneous GHEBE (eq. 17) when $\tau$, $l_h$, $\lambda$, $\underline{\alpha}$ are constant.

By comparing this derivation with that of the homogeneous GHEBE via the classical Laplace-Fourier transform method (section 2.1), it is clear that Babenko's method is very similar, but is more general. Whereas in the homogeneous equation, where the transforms reduce the derivative operations to algebra, the difficulty with Babenko's method is to find proper

interpretations of the fractional operators. However, in the above, we assumed that $\tau$ was only a function of position, so that Laplace (or Fourier) transform methods still apply in the time domain, in the next section we discuss the more challenging interpretation of the fractional inhomogeneous spatial operators.

**3.2 The zeroth order high frequency GHEBE: the HEBE**

Before discussing the inhomogeneous GHEBE, consider the case where the horizontal term $l_h\zeta$ is small compared to $\tau\dfrac{\partial}{\partial t}$;

below we argue that this is a good approximation for scales up to years and decades and greater than tens of kilometers (table 1, appendix A). Alternatively, in globally averaged models, there are no horizontal inhomogeneities so that $\zeta = 0$. In these

cases $\Lambda = \tau\left(\underline{x}\right)^{1/2}\dfrac{\partial^{1/2}}{\partial t^{1/2}}$; and we obtain the inhomogeneous HEBE as a special case of the inhomogeneous FEBE:

$$\tau\left(\underline{x}\right)^{H}{}_{\infty}D_{t}^{H}T_{s}\left(t,\underline{x}\right)+T_{s}\left(t,\underline{x}\right)=\lambda\left(\underline{x}\right)F\left(t,\underline{x}\right); \qquad\qquad H=1/2 \qquad\qquad (32)$$

We have written it with a general $H$ since as in part I, an inhomogeneous version of the EBE may be obtained with $H =$

1. We have also used the Weyl derivative (i.e. from $t = -\infty$) since this accommodated periodic or statistically stationary forcing as well as forcing starting at $t = 0$ (I this case we simply consider $F = 0$ for $t \le 0$). Eq. 32 shows that the HEBE only depends on the local climate sensitivity and the local relaxation time. We'll see below that explicit dependence on the horizontal transport ($v$, $\kappa_h$) and specific heat per volume $\rho c$ is only important at scales somewhat smaller than the transport length scale (or alternatively at extremely long time scales, section 3.5). Before solving the

HEBE, it is instructive to introduce the notation $T_{\infty}\left(t,\underline{x}\right)=\lambda\left(\underline{x}\right)F\left(t,\underline{x}\right)$. $T_{\infty}$ is the equilibrium temperature that would be reached at time $t$, if at each location $\underline{x}$, $F$ was suddenly stopped and fixed at that value. With this notation, we may integrate both sides of eq. 32 by order $H$, and multiply by $\tau^{-H}$ to obtain:

$$T_{s}\left(t,\underline{x}\right)=\frac{1}{\Gamma\left(H\right)}\int_{-\infty}^{t}\left(\frac{t-s}{\tau\left(\underline{x}\right)}\right)^{H-1}\left(T_{\infty}\left(s,\underline{x}\right)-T_{s}\left(s,\underline{x}\right)\right)\frac{ds}{\tau\left(\underline{x}\right)}; \quad 0<H<1 \qquad\qquad (33)$$





Written in this form, it is obvious that the temperature is constantly relaxing in a power law manner to $T_\infty$ (although if $F$ and

is time dependent, equilibrium will in general never in fact be established). In the usual EBM special case ($H = 1$), the power

law must be replaced by an exponential, the HEBE is obtained with $H = 1/2$. Since $T_\infty = \lambda F$, physically the deviation from

$T_\infty$ - the term $\tau^H {}_{-\infty}D_t^H T_s$ (eq. 32) - corresponds to the energy imbalance, as before, it is a power law, long memory energy

storage term.

The FEBE is a linear differential equation that can be solved using Green's functions [*Miller and Ross*, 1993], [*Podlubny*,

1999]. The solution is:

$$T_s\left(\underline{x},t\right)=\frac{\lambda\left(\underline{x}\right)}{\tau\left(\underline{x}\right)}\int_{-\infty}^{t}G_{0,H}\left(\frac{t-s}{\tau\left(\underline{x}\right)}\right)F\left(\underline{x},s\right)ds \qquad (34)$$

where $G_{0,H}$ is the $H$ order Mittag-Leffler impulse response Green's function ([*Lovejoy*, 2019a]). In general, $G_{0,H}$ is only

expressible in terms of infinite series, exceptions are the EBE ($G_{0,1} = e^{-t}$); and the HEBE with in the notation above

$$G_{0,1/2}\left(t\right)=G_{\delta}\left(t;0\right)=\frac{1}{\sqrt{\pi t}}-e^{t}erfc\sqrt{t} \text{ (eq. 31, part I).}$$

The corresponding step response $G_{1,1/2} = G_\Theta$ is the integral of $G_{0,1/2}$ (eq. 32, part I), it describes relaxation to thermodynamic

equilibrium when $F$ is a step function; similarly, the ramp (linear forcing) response $G_{2,1/2}$ (eq. 32, part I), is the integral of the

step response.

**3.3 Some features of stochastic forcing**

The FEBE and the HEBE are examples of fractional relaxation equations; these have primarily been discussed in the context

of deterministic forcings that start at $t = 0$. The corresponding stochastic fractional relaxation processes (in physics, "fractional

Langevin equations", (FLE) see the references in [*Lovejoy*, 2019a]) - here corresponds to stochastic internal forcing. The FLE

have received little attention, although [*Kobelev and Romanov*, 2000], [*West et al.*, 2003] discuss the corresponding

nonstationary random walks. The statistically stationary stochastic case that results when Weyl rather than Riemann-Liouville

fractional derivatives are used is treated in [*Lovejoy*, 2019a], including the HEBE autocorrelation function and prediction

problem (and its limits) when $F$ is a Gaussian white noise.

To understand the noise driven HEBE, it is helpful to Fourier analyze it using $\left({}_{-\infty}D_t^H\right)\overset{Fourier}{\rightarrow}\left(i\omega\right)^H$ [*Lovejoy*, 2019a], section

3.3 part I and appendix C. At high frequencies, the derivative (energy storage) term dominates so that the temperature is a





fractional integral (order $H$) of the forcing. At low frequencies, the derivative term can be neglected so that $T \approx \lambda F$ implying

that the equilibrium temperature follows the forcing and that $\lambda$ is indeed the usual climate sensitivity.

Alternatively, in real space, if $F(t)$ is a unit step function $\Theta(t)$ and $\lambda = 1$, then for $H \neq 1$ the long time relaxation to the

equilibrium temperature response, is a power law: $G_{\Theta,H}(t) = G_{1,H}(t) \approx 1 - t^{-H}$ (part I eq. 33). Similarly, for small $t$, for $H$

$< 1$, the impulse response is singular $G_{0,H}(t) \approx t^{H-1}$ (part I eq. 33). Due to this singularity, when $F(t)$ is a Gaussian white

noise, at high frequencies, $T$ will be a fractional Gaussian noise (fGn) with exponent $H_{fGn} = H - \frac{1}{2}$; averages over time $\Delta t$ will

behave as $\left\langle T_{\Delta t}^2 \right\rangle^{1/2} \propto \Delta t^{H_{fGn}}$. When $H \leq 1/2$ ($H_{fGn} \leq 0$) this implies strong resolution dependencies (mathematically,

divergences) when the resolution is increased ($\Delta t \to 0$) and so is important in data analysis, including the estimation of the

temperature of the Earth [*Lovejoy*, 2017]. When forced by a white noise, the HEBE is exactly at the critical value $H_{fGn} = 0$

corresponding to a "1/f" noise (note that the Earth's internal variability forcing is not necessarily a white noise, it might have

a different scaling behaviour). A particularly relevant aspect is that the correlation function and spectrum change very slowly

from high to low frequencies [*Lovejoy*, 2019a]. With data over a limited ranges of scales – e.g. months to decades – then,

depending on the relaxation time $\tau$, the HEBE could mimic the FEBE with any $H$ in the range $0 < H \leq \frac{1}{2}$ (hence $-1/2 \leq H_{fGn} \leq 0$).

It can therefore potentially account for the geographical variations in $H$ reported in [*Lovejoy et al.*, 2017] as being spurious

consequences of geographical variations in $\tau(\underline{x})$.

At global scales, the high and low frequency HEBE behaviours are close to observations. For example, the global value $H =$

$0.5 \pm 0.2$ was found for the long time behaviour needed to project the earth's temperature to 2100 [*Hebert*, 2017]; the value $H$

$= 0.42 \pm 0.03$ for the internal macroweather variability needed to make monthly and seasonal forecasts [*Del Rio Amador and*

*Lovejoy*, 2019] (note that this was inferred by make the usual assumption that the internal forcing $F$ is a Gaussian white noise,

and this may not be the case). Appendix B discusses the spatial cross correlation matrix implied by the HEBE that is needed

for example in calculating Empirical Orthogonal Functions (EOFs).

We could also mention that if $F$ is spatially statistically homogeneous and independent of the parameters $\lambda$, $\tau$, then not only

will the macroweather temperature fluctuations be well reproduced, but also, up to the relaxation time, the temperature may

easily respect a space-time symmetry called space-time statistical factorization, ("STSF"; e.g.

$R_{space-time}(\Delta\underline{x}, \Delta t) = R_{space}(\Delta\underline{x}) R_{time}(\Delta t)$ where $R$ represents the autocorrelation function), see appendix C. Empirically, the STSF

is at least approximately obeyed by space-time temperature and precipitation fluctuations ([Lovejoy and de Lima, 2015]), and

if respected, the STSF has important implications for macroweather temperature forecasting.

Although the HEBE was derived for anomalies, these were not defined as small perturbations but rather as time-varying

components of the full solution of the temperature (energy) equation with the time independent part corresponding to the

climate state. The only point at which $T$ was assumed to be small was with respect to the absolute local climate temperature

about which the black body radiation was linearized, a fairly weak restriction on $T$. We could also mention that by allowing



the albedo or other parameters to change in time, the HEBE could easily be extended to the study of past or future climates
where it would broaden the spectrum potentially improving the modeling of glacial cycles.

An important feature of fractional differential operators is that they imply long memories, this is the source of the skill in
macroweather forecasts ([*Lovejoy et al.*, 2015], [*Del Rio Amador and Lovejoy*, 2019]). The fractional term with the long
memory corresponds to the energy storage process. In contrast, [*Lionel et al.*, 2014] introduced a class of ad hoc Energy

Balance Models with Memory (EBMM) whose (nonfractional) time derivative depends on integrals over the past state of the
system.

**3.4 The first order in space GHEBE**

The HEBE is the GHEBE limit where horizontal transport effects are dominated by temporal relaxation processes and are

ignored. Although this spatial scale depends on the time scale, appendix A estimates that at monthly time scales, this spatial
scale is of the order of ≈1 km and even at centennial scales it may only be only 100km or so. For these small spatial scales,
we follow [*Babenko*, 1986], [*Kulish and Lage*, 2000], [*Magin et al.*, 2004], and expand the square root operator using the
binomial expansion:

$$\Lambda = \tau^{1/2}\sqrt{\frac{\partial}{\partial t}+V\zeta} \approx \left(\tau\frac{\partial}{\partial t}\right)^{1/2}\left(1+\frac{1}{2}\left(\frac{\partial}{\partial t}\right)^{-1}V\zeta-\frac{1}{8}\left(\frac{\partial}{\partial t}\right)^{-2}\left(V\zeta\right)^2+...\right)$$

$$V = \frac{l_h}{\tau} = \left(\frac{\kappa_h}{\kappa_v}\right)\frac{1}{\rho c\lambda} \qquad\qquad (35)$$

As usual with Babenko's method, a rigorous mathematical justification is not available ([*Podlubny*, 1999]), although recall
that τ, and $l_h$ are only functions of position so that for the temporal operator, Laplace and Fourier transforms techniques still
work.

Considering the spatial part of the fractional operator, we see that it is weighted by the effective heat transport velocity $V$; as

shown below, it plays the role of a small parameter (table 1, appendix A estimate it as ≈10⁻⁴m/s). Therefore, dropping the
subscript "*s*" here and below, the GHEBE is:

$$\tau^{1/2}\left(\frac{\partial}{\partial t}+V\zeta\right)^{1/2}T+T=$$

$$\tau^{1/2}\,_{-\infty}D_t^{1/2}T+T+\frac{1}{2}V\tau^{1/2}\left(_{-\infty}D_t^{-1/2}\zeta\right)T-\frac{1}{8}V^2\tau^{1/2}\left(_{-\infty}D_t^{-3/2}\zeta^2\right)T+...=\lambda F$$

$$(36)$$





with the Weyl fractional derivatives (these are partial fractional derivatives).

Keeping only the spatial terms leading in the small parameter $V$, we have the first order (in space) GHEBE:

$$\tau^{1/2} {}_{-\infty}D_t^{1/2}T + T + \frac{1}{2}V\tau^{1/2}\left({}_{-\infty}D_t^{-1/2}\zeta\right)T = \lambda F \tag{37}$$

Or:

$$\tau^{1/2} {}_{-\infty}D_t^{1/2}T + T + \frac{1}{2}\tau^{1/2} {}_{-\infty}D_t^{-1/2}\left(\underline{v}\cdot\nabla_h T - \kappa_h\nabla_h^2 T\right) = \lambda F \tag{38}$$

This equation is apparently similar to the usual transport equation. To see this, operate on both sides by $\tau^{-1/2} {}_{-\infty}D_t^{1/2}$, to obtain:

$$\frac{\partial T}{\partial t} + \underline{v}'\cdot\nabla T - \kappa'\nabla^2 T + \tau^{-1/2} {}_{-\infty}D_t^{1/2}T = \lambda F' \tag{39}$$

$$\underline{v}' = \frac{1}{2}\underline{v}; \qquad \kappa' = \frac{1}{2}\kappa; \qquad F' = \tau^{-1/2} {}_{-\infty}D_t^{1/2}F$$

Except for the factor ½, the half order derivative term and the "effective", (roughened) forcing, this is the usual transport equation. Nevertheless, although tempting, it would be wrong to think of this simply as a usual transport equation with an extra fractional term. The reason is that the extra term is not a small perturbation, it is dominant except at small spatial scales.

On the contrary, it is rather the classical transport terms that are small perturbations to the main HEBE. Alternatively, without the $\frac{\partial T}{\partial t}$ term, eq. 41 is a generalized fractional diffusion equation (e.g. [*Coffey et al.*, 2012]), although still with a key difference being that the fractional derivative is Weyl, not Riemann-Liouville (i.e. over the range $-\infty$ to $t$, not 0 to $t$).

**3.5 Climate states, Thermodynamic equilibrium and the low frequency GHEBE**

**3.5.1 The HEBE thermodynamic equilibrium**

The HEBE applies to time scales sufficiently short and to spatial scales sufficiently large that the horizontal temperature fluxes are too slow to be important, they are neglected. The first order correction (eqs. 38, 39) makes a small improvement by giving a more realistic treatment of the small scale horizontal transport. However, a long time after performing a step increase of the forcing, the time derivatives vanish and a new climate state is reached. If the temperature followed the pure HEBE, the spatial pattern for thermodynamic equilibrium would be determined by setting the HEBE time derivative to zero:



$$T_{c,HEBE}(\underline{x}) = F_0 \lambda(\underline{x}); \qquad F(t,\underline{x}) = F_0 \Theta(t)$$ (40)

Where the subscript "$c$" indicates the long time (climate) FEBE limit. However, appendix A shows that – depending on the nature of the horizontal transport - at scales perhaps of the order of millennia, the horizontal heat fluxes will dominate the relaxation processes so that for very long times, this HEBE estimate is only approximate.

**3.5.2 Equilibrium and approach to equilibrium in the inhomogeneous GHEBE**

To understand the long time behaviour, we return to the GHEBE but perform a (long-time) binomial expansion of the half-order operator assuming that the transport terms dominate:

$$\left( l(\underline{x})\zeta(\underline{x}) + \tau(\underline{x})\frac{\partial}{\partial t} \right)^{1/2} T = (l\zeta)^{1/2} \left( 1 + (l\zeta)^{-1}\tau\frac{\partial}{\partial t} \right)^{1/2} T$$
$$\approx (l\zeta)^{1/2} T + \frac{1}{2}\frac{\partial}{\partial t}\left( (l\zeta)^{-1/2}\tau \right)T - \frac{1}{8}\frac{\partial^2}{\partial t^2}\left( (l\zeta)^{-1/2}\tau(l\zeta)^{-1}\tau \right)T + ...$$

(41)

(from here on we drop the "$h$" subscripts on $l$ and the gradient operator). We have to be careful since the advection length
and relaxation times are functions of position (but not time) so that the spatial operators don't commute. Keeping terms to first order in time, we obtain:

$$(l\zeta)^{1/2} T + T + \frac{1}{2}\frac{\partial}{\partial t}\left( (l\zeta)^{-1/2}\tau \right)T = \lambda F$$

(42)

To make progess, let's choose the transport operator so that its half powers are easy to interpret. The simplest approach is consider only diffusive transport and to use an isotropic fractional operator defined over the surface of the earth. For an
arbitrary test function ρ, the corresponding order $H$ fractional integral is:

$$\left( -\nabla^2 \right)^{-H/2} \rho = I_{iso,d}^H \rho = \frac{1}{\Gamma(H)} \int_\Omega \frac{\rho(\underline{y}) d^d \underline{y}}{|\underline{x} - \underline{y}|^{d-H}}$$ (43)

(for $0 \leq H \leq d$, where $d$ is the dimension of space, here $d = 2$, see e.g. [*Schertzer and Lovejoy*, 1987], appendix A). This can be understood since in Fourier space, the Laplacian is $-\nabla^2 \xrightarrow{F.T.} |\underline{k}|^2$ and its inverse is $\left( -\nabla^2 \right)^{-1} \xrightarrow{F.T.} |\underline{k}|^{-2}$, the "Poisson solver". Note that eqs. 42, 43 involve ½ order inverse Laplacians which are $H = 1$ (rather than $H = \frac{1}{2}$) isotropic integrals (eq. 43).





With the help of spherical harmonics, Appendix D gives the corresponding operators and their fractional extensions on the surface of the sphere.

Applying eq. 43 to the case $d = 2$ and $H = 1$ we have:

$$\left(-\nabla^2\right)^{-1/2} \rho = \int_\Omega \frac{\rho\left(\underline{x}'\right)d^2\underline{x}'}{\left|\underline{x}-\underline{x}'\right|} \tag{44}$$

Therefore, let us define a diffusive type transport operator $l\zeta$ and its inverse $\left(l\zeta\right)^{-1}$ implicitly from its inverse half-order

power:

$$\left(l\zeta\right)^{-1/2} = l^{-1}\left(-\nabla^2\right)^{-1/2}; \qquad \left(l\zeta\right)^{1/2} = \left(-\nabla^2\right)^{1/2} l = \left(-\nabla^2\right)^{-1/2}\left(-\nabla^2\right)l \tag{45}$$

Hence let us define the half-order operator by:

$$\left(l\zeta\right)^{-1/2} T\left(\underline{x}\right) = l\left(\underline{x}\right)^{-1} \int_\Omega \frac{T\left(\underline{x}'\right)d^2\underline{x}'}{\left|\underline{x}-\underline{x}'\right|} \tag{46}$$

With this definition the surface temperature equation becomes:

$$\frac{1}{2}\frac{\partial}{\partial t}\left[l\left(\underline{x}\right)^{-1}\int_E \frac{\tau\left(\underline{x}'\right)T\left(\underline{x}',t\right)d^2\underline{x}'}{\left|\underline{x}-\underline{x}'\right|}\right] + T\left(\underline{x},t\right) - \int_E \frac{\nabla^2\left(l\left(\underline{x}'\right)T\left(\underline{x}',t\right)\right)d^2\underline{x}'}{\left|\underline{x}-\underline{x}'\right|} = \lambda\left(\underline{x}\right)F\left(\underline{x},t\right) \tag{47}$$

Where the range of the integration $\Omega = E$ is the entire surface of the earth. This equation has only superficial links to equations studied in the literature such as the "generalized fractional advection-dispersion equation" (e.g. [*Meerschaert and Sikorskii*, 2012], [*Hilfer*, 2000]). We can now consider the system reaching equilibrium after a step forcing $F(\underline{x},t) = F_0(\underline{x})\Theta(t)$, (increase by $F_0(\underline{x})$ "turned on" at $t = 0$). At long enough times, the earth reaches thermodynamic equilibrium and the time derivative

term vanishes and we obtain the equation for the climatological temperatures:

$$T_c\left(\underline{x}\right) - \int_E \frac{\nabla^2\left(l\left(\underline{x}'\right)T_c\left(\underline{x}'\right)\right)d^2\underline{x}'}{\left|\underline{x}-\underline{x}'\right|} = \lambda\left(\underline{x}\right)F_0\left(\underline{x}\right) \tag{48}$$

To obtain an approximate solution, let's now assume that $T_c\left(\underline{x}\right)$ differs from the climatological FEBE climate temperature $T_{c,FEBE}(\underline{x})$ by a small perturbation $\delta T(\underline{x})$.

$$T_c\left(\underline{x}\right) = T_{c,HEBE}\left(\underline{x}\right) + \delta T\left(\underline{x}\right); \quad T_{c,HEBE}\left(\underline{x}\right) = \lambda\left(\underline{x}\right)F_0\left(\underline{x}\right) \tag{49}$$





then, using $T_c(\underline{x}) \approx \lambda(\underline{x}) F_0(\underline{x})$ in the integral, we obtain the approximation:

$$T_c(\underline{x}) \approx T_{c,HEBE}(\underline{x}) + \delta T(\underline{x}); \qquad \delta T(\underline{x}) = \int_E \frac{\nabla^2 \left( l(\underline{x}') \lambda(\underline{x}') F_0(\underline{x}') \right) d^2 \underline{x}'}{|\underline{x} - \underline{x}'|} \qquad (50)$$

$\delta T(\underline{x})$ is the slow, diffusive correction to the "instantaneous" (fast, high frequency), HEBE climate sensitivity $\lambda(\underline{x})$ that is estimated at usual (e.g. decadal) scales. As expected, since this is the long time solution after a step perturbation, it doesn't depend on $\tau$.

Horizontal transport of heat redistributes the energy fluxes locally, but since the GHEBE is linear, it shouldn't affect the overall (global) energy balance. Let us check this by direct calculation of the globally averaged temperature. Averaging eq. 48, we obtain:

$$\overline{T_c(\underline{x})} - \overline{\int_E \frac{\nabla^2 \left( l(\underline{x}') T_c(\underline{x}') \right) d^2 \underline{x}'}{|\underline{x} - \underline{x}'|}} = \overline{\lambda(\underline{x}) F_0(\underline{x})}; \qquad \begin{aligned} \overline{f} &= \frac{1}{A_E} \int_E f(\underline{x}) d^2 \underline{x} \\ A_E &= \int_E d^2 \underline{x} \end{aligned} \qquad (51)$$

Where the spatial averaging operator (overbar) is defined for an arbitrary function $f$. The average of the horizontal heat flux term yields:

$$\frac{1}{A_E} \int_E \frac{d^2 \underline{x}}{|\underline{x} - \underline{x}'|} \nabla^2 \left( l(\underline{x}') T_c(\underline{x}') \right) d^2 \underline{x}' = K_E \int_E \nabla^2 \left( l(\underline{x}') T_c(\underline{x}') \right) d^2 \underline{x}' = \int_{\delta E} d\underline{s} \cdot \nabla \left( l(\underline{x}') T_c(\underline{x}') \right) = 0 \qquad (52)$$

Where $K_E$ is an unimportant constant from the $\underline{x}$ integration, independent of $\underline{x}'$. The far right equality is an application of the divergence theorem on the surface $E$ whose boundary is $\delta E$, $d\underline{s}$ is a vector parallel to the bounding line. But since the integration is over the whole earth surface ($E$), there is no boundary, hence the result. We conclude that while horizontal diffusion transports heat over the earth's surface, it does not affect the overall global radiation budget: $\overline{T_c} = \overline{T_{c,FEBE}}$.

## 4. Conclusions

Up until now, at macroweather and climate scales, the Earth's energy balance has been modelled using two classical approaches. On the one hand, Budyko - Sellers models assume the continuum mechanics heat equation holds, this yields a 1-D latitudinally varying climate state. On the other hand, there are the zero-dimensional box models that combine Newton's law of cooling with the assumption of an instantaneous temperature-storage relationship. Both models avoid the critical





conductive - radiative surface boundary conditions; the former by ignoring heat storage, redirecting radiative imbalances meridionally away from the equator, the latter by postulating a surface heat flux that is not simultaneously consistent with the heat equation and energy conservation across a conducting and radiating surface (part I).

This two part paper re-examined the classical heat equation with classical semi-infinite geometry. In the horizontally homogeneous case (part I), the novelty is the treatment of the conductive - radiative boundary conditions, here (part II), it is the use of Babenko's method to extend this to the more realistic horizontally inhomogeneous problem. In both cases, the semi-infinite subsurface geometry is only important over a shallow layer of the order of the diffusion depth where most of the storage occurs (roughly estimated as $\approx 100$m in the ocean, $\approx <10$m over land, see table 1 and appendix A).

The key result was obtained by using standard Laplace and Fourier techniques. It was shown quite generally that the surface temperatures and heat fluxes are related by a half-order derivative relationship. This means that if Budyko-Sellers models are right - that the continuum mechanics heat equation is a good approximation to the Earth averaged over a long enough time – then a consequence is that the energy stored is given by a power law convolution over its past history. This is a general consequence of the conductive - radiative surface boundary condition and is very different from the box models that assume that the relationship between the temperature and heat storage is instantaneous.

If we ignore horizontal heat transport (part I), an immediate consequence of half order storage is that the temperature obeys the Half-order Energy Balance Equation (HEBE) rather than the classical order one EBE. The implied long time storage behaviour explains the success of scaling based climate projections and, the implied short time behaviour potentially explains the success of macroweather forecasts that exploit it. When the system is periodically forced, the response is shifted in phase - and borrowing from the engineering literature - the surface is characterized by a complex thermal impedance that we showed is equal to the (complex) climate sensitivity. In part I, we gave evidence that this quantitatively explains the phase lag (typically of about 25 days) between the annual solar forcing and temperature response.

In this second part, we investigated the consequences of horizontal heat transport, first in a homogeneous medium with inhomogeneous forcing (section 2) and then, more generally with inhomogeneous material properties (including variable diffusion lengths, relaxation times, and climate sensitivities, section 3). Laplace and Fourier techniques and not so useful here, but the extension to inhomogeneous media was nevertheless possible thanks to Babenko's powerful (but less rigorous) operator method. Whereas in part I, the homogeneous fractional space-time operator was given a precise meaning, here - following Babenko - the corresponding inhomogeneous operator was interpreted using binomial expansions for both the short and long time limits and yield 2D energy balance models.

The expansions depend both on the space and time scale and on a dimensional parameter: the typical horizontal transport speed ($V$), estimated as $\approx 10^{-4}$m/s (appendix A). The zeroth order expansion in time limit yielded the inhomogeneous HEBE, the first order correction yielded an equation that superficially resembled the usual heat equation but instead had a leading half - order time derivative term. Based on the analysis of NCEP reanalyses (appendix A), it was argued that at spatial scales larger than hundreds of kilometers, that these approximations are likely to be useful for years, decades, and perhaps longer. However, for studying climate states – defined for example as the thermodynamic equilibrium state for forcings that are increased





everywhere in step function fashion – we required low, not high frequency expansions and these are based on fractional spatial operators.  We defined inhomogeneous fractional diffusion operators in both flat space and on the sphere (appendix D), and derived equations for both the thermodynamic limit and the approach to the limit.  We showed that (as expected) they conserved energy and that the low frequency climate sensitivity is somewhat different from that estimated at higher frequencies (from the EBE or HEBE).

The EBE and HEBE are the $H = 1$, $H = 1/2$ special cases of the Fractional EBE (FEBE) that was recently introduced as a phenomenological model [*Lovejoy*, 2019a], [*Lovejoy*, 2019b] with empirical estimates $H \approx 0.4$ - $0.5$, i.e. very close to the HEBE.  Although only a special case, the HEBE illustrates the general features of the FEBE fractional-order energy storage term and power law long memories.  [*Lovejoy*, 2019a]  discussed the statistical properties of the FEBE driven by Gaussian white noise (a model for the internal variability forcing) showing that the high frequency limit is a process called fractional

Gaussian noise (fGn).  In the special HEBE case with $H = 1/2$, the fGn temperature response has exactly a high frequency $1/f$ spectrum that is cut-off at the relaxation time (empirically of the order of a few years).  [*Lovejoy*, 2019a] developed optimal predictors and determined the predictability skill.

Whereas the more general FEBE is essentially a phenomenological model up unitl now justified by the hypothesized scale invariance of the energy storage mechanisms, the HEBE follows directly and quite generally from the continuum mechanics

heat equation, thus giving it a more solid theoretical basis.  However, the work here suggests another way to obtain the FEBE: to replace the classical heat equation by its fractional generalization, a possibility that we explore elsewhere.  Part II allowed us for the first time to extend energy balance models to 2-D, allowing the treatment of regional temporal anomalies.  Depending on the space-time statistics of the anomaly forcing, the HEBE justifies the current Fractional EBE (FEBE) based macroweather (monthly, seasonal) temperature forecasts [*Lovejoy et al.*, 2015], [*Del Rio Amador and Lovejoy*, 2019].  Similarly, the low

frequency (asymptotic) power law part can produce climate projections with significantly lower uncertainties than current GCM based alternatives ([*Hebert*, 2017] and work in progress directly using the HEBE with R. Procyk).

This work was performed in the spirit of Budyko-Sellers models in which the Earth system is averaged over scales longer than typical lifetimes of planetary scale weather structures.  Following Budyko-Sellers, the key physical assumption was that the resulting averaged system is a continuum system, thus justifying use of the general continuum mechanics heat equation.  From

this, the GHEBE and HEBE follow from the surface conductive-radiative boundary condition.  In as much as GCMs (that are based on continuum mechanics) reproduce the same statistics as the noise – or anthropogenically forced FEBE and HEBE, the continuum hypothesis is plausible.

As a final comment, we should mention that although this paper focused on the time varying anomalies with respect to a time independent climate state, our approach opens the door to new methods for determining full 2-D climate states

(generalizations of the 1-D Budyko-Sellers type climates) but also to determining past and future climates and the transitions between them.  This is because the definition of temperature "anomalies" is very flexible.  For example, we could first apply the method to determining the existing climate by fixing the forcing at current values and solving the time independent transport equations.  Then, the long term effect of changes such as step function increases in forcing could be determined from the





GHEBE anomaly equation (section 3.5) which regionally corrects the local climate sensitivities for (slow) horizontal energy transport effects. Nonlinear effects that can be modelled by temperature dependent forcings (i.e. $F(\underline{x},t) \rightarrow F(\underline{x},t,T(\underline{x},t))$) can easily be introduced. Other nonlinear effects needed to account for Milankovitch cycles could thus easily be made, the primary difference being the half-order derivatives and the scaling that they imply. Indeed, the power law relaxation processes implied by the GHEBE suggests straightforward explanations for the observed power law climate regime spanning the range from centennial to Milankovitch scales.

## 5. Acknowledgements

I acknowledge discussions with L. Del Rio Amador, R. Procyk, R. Hébert, D. Clarke and C. Penland. This is a contribution to fundamental science; it was unfunded and there were no conflicts of interest. The data used in appendix A are from the NOAA website: https://www.esrl.noaa.gov/psd/data/gridded/data.ncep.reanalysis.html.

### Appendix A: Empirical analysis of the horizontal structure

In order to apply our results to the Earth, we need some idea of the magnitudes of various terms in our equations. To start with, recall that our model is of the Earth system at macroweather and climate time scales i.e. all relevant quantities are averaged over the weather scales $\approx 10$ days or longer. The resulting averaged system is then treated as a continuum and the general continuum mechanics heat equation is applied. In this, we essentially follow the Budyko-Sellers approach and consider that the diffusive transport is characterized by eddy (not molecular) diffusivities and that the vertical structure of this averaged continuum is homogeneous (although it may vary considerably from place to place in the horizontal). Unlike Budyko-Sellers that treat the vertical as negligibly thick – they don't consider it at all – our key difference is that we assume that it has a thickness of the order of a few diffusion depths, and then we apply the key conductive- radiative surface boundary condition.

Probably the most important aspect is to estimate the relative importance of the temporal relaxation (and storage) terms $\tau \partial / \partial t$ in comparison to the horizontal transport terms $l_h \zeta$ (see eq. 25). Indeed, for judging their relative importance, the key parameter is the ratio of the transport to relaxation terms $r$:

$$r = V \frac{\zeta T}{(\partial T / \partial t)}; \qquad V = \frac{l_h}{\tau}; \qquad \alpha = \frac{v}{V} \qquad (53)$$





Where $\alpha$ is the magnitude of the dimensionless advection velocity vector $\underline{\alpha} = \underline{v}/V$. When $r \ll 1$, the transport term is small compared to the temporal term, conversely when $r \gg 1$. In order to quantify this, it is convenient to consider the advective ("$a$") and diffusive ("$d$") terms as well as their derivatives individually:

$$r_a = V \frac{\zeta_{a,x}T + \zeta_{a,y}T}{(\partial T / \partial t)}; \qquad \zeta_{a,x}T \approx \alpha_x \frac{\partial T}{\partial x}; \qquad \zeta_{a,y}T \approx \alpha_y \frac{\partial T}{\partial y}$$

(54)

$$r_d = V \frac{\zeta_{d,x}T + \zeta_{d,y}T}{(\partial T / \partial t)}; \qquad \zeta_{d,x}T = l_h \frac{\partial^2 T}{\partial x^2}; \qquad \zeta_{d,y}T = l_h \frac{\partial^2 T}{\partial y^2}$$

In the macroweather regime, the temporal temperature fluctuation at time scale $\Delta t$ is $\Delta T(\Delta t) \approx T_{\Delta t}$ where $T_{\Delta t}$ is the anomaly average over scale $\Delta t$; empirically this is valid over the macroweather regime i.e. up to 10 - 30 years in the industrial epoch (see e.g. [*Lovejoy and Schertzer*, 2013], [*Lovejoy*, 2013], [*Lovejoy et al.*, 2017]). The typical fluctuation can be estimated by the RMS anomaly:

$$s_{\Delta t}(\underline{x}) = \left(\overline{T_{\Delta t}^2}\right)^{1/2} \approx s_1(\underline{x}) \left(\frac{\Delta t}{\Delta t_1}\right)^{H_t}$$

(55)

Where the overbar is the average over all the anomalies in a time series at a single location $\underline{x}$. $\Delta t_1$ is convenient reference time, here taken as 1 month. Empirically, the exponent $H_t \approx 0$ to -0.2; this similar to the high frequency result $H_t = 0$ (i.e. for $\Delta t < \tau$) predicted from the HEBE with white noise forcing, valid for $\Delta t \approx < \tau$. Hence for our present purposes the typical time derivative is:

$$\frac{\partial T}{\partial t} \approx \frac{s_{\Delta t}}{\Delta t}$$

(56)

This is the resolution $\Delta t$ time derivative. Since typical north-south gradients are larger than typical east-west ones, the meridional ($y$) component is dominant, so that we will focus on it:





$$\frac{\partial T}{\partial y} \approx \frac{\left(\overline{\Delta T_{\Delta t}\left(\Delta y\right)^2}\right)^{1/2}}{\Delta y} = \frac{\Delta s_{\Delta t}\left(\Delta y\right)}{\Delta y}; \qquad \frac{\partial^2 T}{\partial y^2} \approx \frac{\Delta\left(\overline{\Delta T_{\Delta t}\left(\Delta y\right)^2}\right)^{1/2}}{\Delta y^2} = \frac{\Delta^2 s_{\Delta t}\left(\Delta y\right)}{\Delta y^2}$$

(57)

Hence:

$$r_{a,y} = V\alpha \frac{\Delta t}{\Delta y} \Delta \log s_{\Delta t}\left(\Delta y\right)$$

(58)

$$r_{d,y} = Vl_h \frac{\Delta t}{\Delta y^2}\left(\left(\Delta \log s_{\Delta t}\left(\Delta y\right)\right)^2 + \Delta^2 \log s_{\Delta t}\left(\Delta y\right)\right)$$

Where $\Delta \log s_{\Delta t}\left(\Delta y\right) = \dfrac{\Delta s_{\Delta t}\left(\Delta y\right)}{s_{\Delta t}}$, is the relative fluctuation in the RMS temperature at time scale $\Delta t$, spatial scale $\Delta y$ and

- since we are only interested in an order of magnitude - we took $\alpha \approx \alpha_y$. The estimate of the diffusive term uses a finite difference approximation to the Laplacian. $l_h$ is horizontal anomaly relaxation length and $\alpha$ is the nondimensional advection speed $v/V$ ($V = l_h/\tau$, see below). To gauge the order of magnitudes, in the far right of eq. 58, we suppressed the signs.

To estimate $l_h$, consider the volumetric specific heat $\rho c$. Ocean and land values are similar (respectively water: $\rho c \approx 4\times10^6$ and soil: $\rho c \approx 1\times10^6$ J/m$^3$). For $\lambda$, the global mean value is $\approx 0.8\pm0.4$ K/W/m$^2$, (using the $CO_2$ doubling value 3±1.5C, 90% confidence interval and 3.71 W/m$^2$ for $CO_2$ doubling) with regional values a factor of $\approx 2$ higher or lower (IPCC AR5) yielding $\rho c\lambda \approx 3\times10^6$ s/m. The horizontal (eddy) diffusivity is $\kappa_h \approx 1$ m$^2$/s ([*Sellers*, 1969], [*North et al.*, 1981]). The vertical diffusivity is not used in the usual energy balance models, however in climate models, ocean values of $\kappa_v \approx 10^{-4}$ m$^2$/s are typical [*Houghton et al.*, 2001]. For soil, rough values of $\kappa_v \approx 10^{-6}$ m$^2$/s (wet) and $\kappa_v \approx 10^{-7}$ m$^2$/s (dry) are measured in [*Márquez et al.*, 2016] so that for soils, $l_v \approx 3 - 10$m.

Alternatively we can use $\kappa_v = \tau/(\rho c\lambda)^2$ and the global estimates of $\tau \approx 10^8$s ([*Hebert*, 2017], work in progress with R. Procyk, or part I, section 3.3). From these, we obtain $\kappa_v \approx 10^{-5}$ m$^2$/s which is close to the model values. In conclusion, using $\kappa_v \approx 10^{-5} - 10^{-4}$ m$^2$/s yields $l_v \approx 30 - 100$m, $l_h \approx 10$ km. Consequently, the diffusive based velocity parameter is $V \approx l_h/\tau \approx 10^{-4}$ m/s.

The best transport model – diffusive, advective – or both - is not clear, therefore let us estimate the magnitude of the advective velocity $v$ assuming that it dominates the transport. The appropriate value is not obvious since most models just use eddy diffusivity – not advection - for transport. One way - for example [Warren and Schneider, 1979] - is to note that typical meridional heat fluxes are of the order of 100 W/m$^2$ over meridional bands whose temperature gradients $\Delta T$ are several degrees



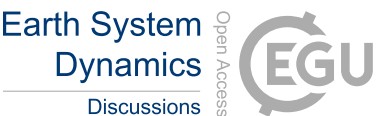

K. If this heat is transported by advection, it implies $v \approx Q_a/(\rho c \Delta T) \approx 10^{-5}$ - $10^{-4}$m/s (eq. 4, part I), hence, using $V \approx 10^{-4}$m/s (above), we find $\alpha = v/V \approx 0.1$ - 1.

With these dimensional and nondimensional parameter estimates, the final step is to estimate values of the gradient and Laplacian terms (eq. 58). Since $s$ - and hence $\log s$ - are the amplitudes of temporal noises; these amplitudes vary stochastically from one spatial location to another. Due to the spatial scaling, we expect that their statistics follow power laws up to large scales. To quantify this, we used NCEP reanalysis data at 2.5° resolution from 1948 to present, and after removing the low frequency anthropogenic trend, we estimated the RMS temperature anomalies at each pixel; $s(\underline{x})$. In fig. 6, we then calculated spatial zonal and meridional fluctuations $\Delta \log s(\Delta x)$, $\Delta \log s(\Delta y)$, and from these their root mean square (RMS) values. From the figure, we see that to a good approximation:

$$\Delta \log s(\Delta x) \approx \left(\frac{\Delta x}{L_{EW}}\right)^{H_x} \qquad L_{EW} \approx 1.5 \times 10^7 \, m$$

$$\Delta \log s(\Delta y) = \left(\frac{\Delta y}{L_{NS}}\right)^{H_y} \qquad L_{NS} \approx 3 \times 10^6 \, m \qquad H_x \approx H_y \approx 0.5$$

(59)

The fluctuations are Haar fluctuations, but because $H_x \approx H_y > 0$, they are nearly equal to difference fluctuations [*Lovejoy and Schertzer*, 2012]. We see that the zonal and meridional lines are roughly parallel: with a "trivial" horizontal anisotropy factor $\approx 5$. Although, $H = 1/2$ is the value corresponding to Brownian motion, the actual variability is highly intermittent (spiky), so that the increments are far from Gaussian; it is *not* Brownian motion. Multifractal analysis indicates that the intermittency parameter (the codimension of the mean) $C_1 \approx 0.16$ which is very high, reflecting the strong spatial fluctuations as we move from one climate zone to another [*Lovejoy*, 2018], [*Lovejoy*, 2019b].

Since the north-south gradients are much stronger than the east-west ones, we can estimate the gradients and Laplacians by using the $y$ direction fluctuations: at scale $\Delta y$:

$$r_{a,y} = \frac{V \alpha \Delta t}{\Delta y}\left(\frac{\Delta y}{L_{NS}}\right)^{H_y} \qquad (60)$$

$$r_{d,y} = \frac{V \Delta t}{\Delta y}\left(\frac{l_h}{\Delta y}\right)\left[\left(\frac{\Delta y}{L_{NS}}\right)^{2H_y} + \left(\frac{\Delta y}{L_{NS}}\right)^{H_y}\right] \qquad (61)$$





Since $L_{NS} \approx 3 \times 10^6 \text{m}$ , over most of the range of $\Delta y$, $r_{d,y} \approx \dfrac{V \Delta t}{\Delta y} \left( \dfrac{l_h}{\Delta y} \right) \left( \dfrac{\Delta y}{L_{NS}} \right)^{H_y}$ so that the ratio of advection to diffusion is

$\dfrac{r_c}{r_d} \approx \left( \dfrac{\alpha \Delta y}{l_h} \right)$ so that advection dominates diffusion for $\Delta y > \dfrac{l_h}{\alpha}$. Taking $\alpha \approx 1$, it is dominant for $\Delta y \gg \approx l_h$.

Using $l_h \approx 10^4 \text{m}$, $L_{NS} \approx 3 \times 10^6 \text{m}$, $H_y = 1/2$, $V = 10^{-4}$ m/s we find approximately critical length scales that yields unit ratios:

$$\Delta y_{c,a} = 10^{-14} \Delta t^2; \qquad r_a \left( \Delta y_{c,a} \right) = 1$$

$$\Delta y_{c,d} = 10^{-2} \Delta t^{2/3}; \qquad r_d \left( \Delta y_{c,d} \right) = 1$$

(62)

Where $\Delta t$ is measured in seconds, $\Delta y$ in meters. When the typical distances exceed these critical distances (i.e. when $\Delta y > \Delta y_c$), we have $r < 1$ so that the temporal derivative terms dominate over the horizontal transport. For $\Delta t = 1$ month, we have $\Delta y_{c,a} \approx$ 0.1m, and $\Delta y_{c,d} \approx 200$m, so that unless the distances are very small, the temporal (storage) terms are indeed dominant. Even over much longer time scales - e.g. $\Delta t \approx 30$ years $(10^{10}\text{s})$, they dominate for distances greater than $\approx \Delta y_{c,a} \approx \Delta y_{c,d} \approx 10$ km. Alternatively, we could estimate the time scales needed so that the critical transport scale is 1000km. From the same equations,

we obtain estimates of 300 years (advection), 30,000 years (diffusion). Note however that in the anthropocene, for periods $\Delta t \gg > 10$ years, that the temporal fluctuations start to grow (i.e. the empirical relations eqs. 60, 61 will break down); nevertheless, the above scaling relations for the internal variability may hold to much longer times [*Lovejoy et al.*, 2013].

In summary, from eq. 62, we conclude that for the larger scales $>> \approx 10$ km, that $r \ll 1$ and that the HEBE may apply except for time scales $\gg \tau$: the only explicit role of $\kappa_h$, $\kappa_v$, $\rho$, $c$ is to determine the limits of validity of the HEBE via $l_h$, $\alpha$. When the

HEBE is valid, only the relaxation time $\tau$ and the climate sensitivity $\lambda$ are relevant.

**Appendix B: The HEBE cross-correlations**

The temperature anomaly cross-correlation function (a matrix when the temperature is discretized on a grid), is commonly used in climate science, notably to determine Empirical Orthogonal Functions (EOFs). These can be determined from the HEBE (or GHEBE if needed) once a forcing model is given. Let us first consider that the climate sensitivities and relaxation

times are deterministic characterizations of the local properties at points $\underline{x}_1$, $\underline{x}_2$. In this case, for the HEBE, any correlations between the temperature anomalies at those points will arise because of correlations in the forcing $F(\underline{x},t)$. We now consider simple deterministic and stochastic forcings.





**a) Deterministic forcing:**

The simplest model is to take complete spatial correlation correlations, with a step function ($\Theta(t)$) forcing at $t = 0$, but different
at each position $\underline{x}$:

$$F(\underline{x},t) = F_0(\underline{x})\Theta(t) \tag{63}$$

The temporally averaged cross-correlation can be determined by:

$$T(\underline{x}_1,t)T(\underline{x}_2,t) = \frac{\lambda(\underline{x}_1)F_0(\underline{x}_1)\lambda(\underline{x}_2)F_0(\underline{x}_2)}{\tau(\underline{x}_1)\tau(\underline{x}_2)}\int_0^t\int_0^t G_{0,1/2}\left(\frac{t-s_1}{\tau(\underline{x}_1)}\right)G_{0,1/2}\left(\frac{t-s_2}{\tau(\underline{x}_2)}\right)ds_1\,ds_2 \tag{64}$$


Recalling that $G_{1,1/2}$ ($= G_\Theta$) is the step response, the integral of $G_{0,1/2}$ ($= G_\delta$) since $G_{1,1/2}(\infty) = 1$ we have:

$$\lim_{t_L\to\infty}\left[\frac{1}{t_L}\int_0^{t_L}G_{1,1/2}\left(\frac{t}{\tau(\underline{x}_1)}\right)G_{1,1/2}\left(\frac{t}{\tau(\underline{x}_2)}\right)dt\right] = 1 \tag{65}$$

Hence:

$$\overline{T(\underline{x}_1,t)T(\underline{x}_2,t)} = \lambda(\underline{x}_1)F_0(\underline{x}_1)\lambda(\underline{x}_2)F_0(\underline{x}_2) \tag{66}$$

**b) Stochastic forcing:**

A convenient model of pure internal variability, is to assume that the forcing is statistically stationary in time with the following forcing cross-correlations:

$$R_F(\underline{x}_1,\underline{x}_2,\Delta t) = < F(\underline{x}_1,t)F(\underline{x}_2,t-\Delta t) > \tag{67}$$

(the "<>" symbol indicates ensemble, statistical averaging). This implies a stationary temperature cross-correlation:

$$R_T(\underline{x}_1,\underline{x}_2,\Delta t) = < T(\underline{x}_1,t)T(\underline{x}_2,t-\Delta t) > \tag{68}$$



Note the general symmetry property $R\left(\underline{x}_1,\underline{x}_2,-\Delta t\right)=R\left(\underline{x}_2,\underline{x}_1,\Delta t\right)$; we only need to determine $R$ for $\Delta t>0$. For statistically stationary forcing, $R_T\left(\underline{x}_1,\underline{x}_2,\Delta t\right)$ is the anomaly cross-correlation needed - for example - for constructing Empirical Orthogonal Functions (EOFs).

The easiest way to relate $R_F$ and $R_T$ is via their spectra. Let us define the transform pairs:

$$\widehat{T(\omega)}=\int_{-\infty}^{\infty}e^{-i\omega t}T(t)dt; \quad T(t)=\frac{1}{2\pi}\int_{-\infty}^{\infty}e^{i\omega t}\widehat{T(\omega)}d\omega$$


(69)

similarly for the forcing $F$ (the circonflex indicates Fourier Transform). Then:

$$\widehat{\left(\frac{d^H T}{dt^H}\right)}=\left(i\omega\right)^H\hat{T}$$

(70)

(this is true for the Weyl fractional derivatives used here, [*Podlubny*, 1999]). So that the impulse response is:

$$G_{0,1/2}(t)=\frac{1}{2\pi}\int_{-\infty}^{\infty}\frac{e^{i\omega t}}{1+\left(i\omega\right)^{1/2}}d\omega$$

(71)


The solution to the HEBE at two different points $\underline{x}_1$, $\underline{x}_2$ is:

$$\widehat{T^*}\left(\underline{x}_1,\omega_1\right)=\lambda\left(\underline{x}_1\right)\frac{\widehat{F^*}\left(\underline{x}_1,\omega_1\right)}{1+\left(-i\omega_1\tau\left(\underline{x}_1\right)\right)^{1/2}}$$

$$\hat{T}\left(\underline{x}_2,\omega_2\right)=\lambda\left(\underline{x}_2\right)\frac{\hat{F}\left(\underline{x}_2,\omega_2\right)}{1+\left(i\omega_2\tau\left(\underline{x}_2\right)\right)^{1/2}}$$

(72)

Where the asterix indicates complex conjugate. Multiplying and taking ensemble averages and assuming that the forcing – and hence responses - are statistical stationary, we obtain:

$$<\widehat{T^*}\left(\underline{x}_1,\omega\right)\hat{T}\left(\underline{x}_2,\omega'\right)>=R_T\left(\underline{x}_1,\underline{x}_2,\omega\right)\delta\left(\omega-\omega'\right); \qquad R_T\left(\underline{x}_1,\underline{x}_2,\omega\right)=R_T^*\left(\underline{x}_2,\underline{x}_1,\omega\right)$$


(73)

Where:





$$R_T\left(\underline{x}_1,\underline{x}_2,\Delta t\right)=\frac{1}{2\pi}\int_{-\infty}^{\infty}e^{i\omega\Delta t}R_T\left(\underline{x}_1,\underline{x}_2,\omega\right)d\omega$$

(74)

Therefore:

$$R_T\left(\underline{x}_1,\underline{x}_2,\omega\right)=\lambda\left(\underline{x}_1\right)\lambda\left(\underline{x}_2\right)\hat{G}_T\left(\underline{x}_1,\underline{x}_2,\omega\right)R_F\left(\underline{x}_1,\underline{x}_2,\omega\right);$$

615 (75)

$$\hat{G}_T\left(\underline{x}_1,\underline{x}_2,\omega\right)=\frac{1}{\left(1+\left(-i\omega\tau\left(\underline{x}_1\right)\right)^{1/2}\right)\left(1+\left(i\omega\tau\left(\underline{x}_2\right)\right)^{1/2}\right)}$$

A special case that is useful later, is when $\underline{x}_1 = \underline{x}_2 = \underline{x}$, which yields the spectrum $E_T$ at the point $\underline{x}$:

$$E_T\left(\underline{x},\omega\right)\delta\left(\omega-\omega'\right)=\left\langle\hat{T}\left(\underline{x},\omega\right)\widehat{T^*}\left(\underline{x},\omega'\right)\right\rangle;\qquad\qquad E_T\left(\underline{x},\omega\right)=R_T\left(\underline{x},\underline{x},\omega\right)$$

(76)

Using a partial fraction expansion of eq. 75, we obtain:

$$\widehat{G}_T\left(\underline{x}_1,\underline{x}_2,\omega\right)=\frac{1}{\tau_1+\tau_2}\left[\frac{\tau_1+i\tau_g}{\left(1+\left(-i\omega\tau_1\right)^{1/2}\right)}+\frac{\tau_2-i\tau_g}{\left(1+\left(i\omega\tau_2\right)^{1/2}\right)}\right];\qquad\qquad\tau_g=sign\left(\omega\right)\left(\tau_1\tau_2\right)^{1/2}$$

(77)

By inverting the Fourier transform, this can be used to determine the real space transfer function $G_T\left(\underline{x}_1,\underline{x}_2,\Delta t\right)$. Using contour integration, it is convenient to convert the inverse Fourier transforms into Laplace transforms for $\Delta t > 0$:

$$G_T\left(\underline{x}_1,\underline{x}_2,\Delta t\right)=\frac{1}{\pi\left(\tau_1+\tau_2\right)}\left[\int_0^{\infty}e^{-x\left(\Delta t/\tau_2\right)}\frac{x^{1/2}}{1+x}dx+\left(\frac{\tau_2}{\tau_1}\right)^{1/2}\int_0^{\infty}e^{-x\left(\Delta t/\tau_2\right)}\frac{1}{1+x}dx-\left(\frac{\tau_1}{\tau_2}\right)^{1/2}\int_0^{\infty}e^{-x\left(\Delta t/\tau_1\right)}\frac{1}{1+x^{1/2}}dx\right]$$

(78)

For $\Delta t<0$, use $G_T\left(\underline{x}_1,\underline{x}_2,-\Delta t\right)=G_T\left(\underline{x}_2,\underline{x}_1,\Delta t\right)$. The spatial cross-correlation, temporal autocorrelation function of the temperature is therefore:

$$R_T\left(\underline{x}_1,\underline{x}_2,\Delta t\right)=\lambda\left(\underline{x}_1\right)\lambda\left(\underline{x}_2\right)G_T\left(\underline{x}_1,\underline{x}_2,\Delta t\right)*R_F\left(\underline{x}_1,\underline{x}_2,\Delta t\right)$$

(79)





Where the "*" indicates convolution.

The basic Laplace transforms in eq. 78 can be expressed in terms of higher mathematical functions as follows (all for $t>0$):

$$G_{0,1/2}(t) = \frac{1}{\pi} \int_0^\infty \frac{x^{1/2}}{1+x} e^{-xt} dx = \frac{1}{\sqrt{\pi t}} - e^t erfc\left(\sqrt{t}\right)$$

(80)

$$\frac{1}{\pi} \int_0^\infty \frac{e^{-xt}}{1+x} dx = \frac{1}{\pi} e^t \Gamma(0,t); \qquad \Gamma(0,t) = \int_t^\infty \frac{e^{-t}}{t} dt$$

$$\frac{1}{\pi} \int_0^\infty \frac{1}{1+x^{1/2}} e^{-xt} dx = \frac{1}{\sqrt{\pi t}} - e^{-t} erfi\left(\sqrt{t}\right) + \frac{e^{-t}}{\pi} E_I(t); \qquad erfi(z) = erf(iz)/i = \operatorname{Im}\left(ercf(-iz)\right);$$


$$E_I(t) = -\int_{-t}^\infty \frac{e^{-t}}{t} dt = -\Gamma(0,-t) + i\pi$$

The $i\pi$ comes from integrating half way around the pole at the origin. Note that both the Exponential Integral ($E_I$) and the incomplete Gamma functions have log divergences at the origin. If needed, these formulae can be combined to obtain a complete analytic expression for $G_T(\underline{x}_1,\underline{x}_2,\Delta t)$, which can then be used to determine the temperature correlations if the forcing correlations are known: $R_T(\underline{x}_1,\underline{x}_2,\Delta t) = \lambda(\underline{x}_1)\lambda(\underline{x}_2)G_T(\underline{x}_1,\underline{x}_2,\Delta t) * R_F(\underline{x}_1,\underline{x}_2,\Delta t)$ where the asterix is the

temporal convolution.

The special case $\underline{x}_1 = \underline{x}_2$ i.e. with $\tau_1 = \tau_2 = \tau$, is a little simpler:

$$G_T(\Delta t) = \frac{1}{\tau} g\left(\frac{|\Delta t|}{\tau}\right); \qquad g(\Delta t) = \frac{1}{2\pi} \int_0^\infty e^{-x\Delta t} \left(\frac{x^{1/2}}{1+x} + \frac{1}{1+x} - \frac{1}{1+x^{1/2}}\right) dx; \qquad \Delta t > 0$$

(81)

Whose Fourier transform is:

$$\hat{G}_T(\underline{x},\underline{x},\omega) = \frac{1}{1 + 2\operatorname{Re}\left[\left(-i\omega\tau\right)^{1/2}\right] + \omega\tau}$$

(82)

Evaluating the integral for $g(\Delta t)$ using the Laplace transform formulae (eq. 80):

$$g(\Delta t) = \frac{1}{\pi}\left(e^{\Delta t}\Gamma(0,\Delta t) + e^{-\Delta t}\operatorname{Re}\left(\Gamma(0,-\Delta t)\right)\right) - \left(e^{\Delta t}erfc\sqrt{\Delta t} + e^{-\Delta t}\operatorname{Im}\left(erfc\left(-i\sqrt{\Delta t}\right)\right)\right)$$

(83)



($\Delta t > 0$). The small scale and asymptotic limits are thus:

$$g\left(\Delta t\right) = -\frac{\log \Delta t}{\pi} - \frac{1}{2} - \frac{\gamma_E}{\pi} + 2\sqrt{\frac{\Delta t}{\pi}} - \frac{t}{2} - \left(\frac{t^2 \log \Delta t}{2\pi}\right) + ... \qquad\qquad \Delta t << 1$$

$$g\left(\Delta t\right) \approx \frac{1}{\Delta t \sqrt{\pi \Delta t}} - \frac{2}{\pi \Delta t^2} + \frac{15}{8 \Delta t^3 \sqrt{\pi \Delta t}} - ... \qquad\qquad \Delta t >> 1$$

(84)

Note the small scale log divergence, this is important when the forcing is a white noise, see [*Lovejoy*, 2019a].   The temporal autocorrelation at the point $\underline{x}$ is thus:

$$R_T\left(\underline{x}, \Delta t\right) = \frac{\lambda\left(\underline{x}\right)^2}{\tau\left(\underline{x}\right)} g\left(\Delta t / \tau\left(\underline{x}\right)\right) * R_F\left(\underline{x}, \Delta t\right); \qquad\qquad R\left(\underline{x}, \Delta t\right) = R\left(\underline{x}, \underline{x}, \Delta t\right)$$

(85)

However, in general, the Fourier relations are easier to deal with.

**Appendix C: Statistical Space-Time Factorization**

At high frequencies (i.e. $\Delta t < \tau$), and empirically over the macroweather regime up to a decade or more ([Lovejoy and de Lima, 2015]), both precipitation and temperature anomalies (at least approximately) respect a space-time symmetry called "space-time statistical factorization" ("STSF").    For example, for the autocorrelation function $R$, this implies $R_{space-time}\left(\underline{\Delta x}, \Delta t\right) = R_{space}\left(\underline{\Delta x}\right) R_{time}\left(\Delta t\right)$. If obeyed, this factorization implies important simplications in regional macroweather forecasting: it is therefore interesting to investigate the implications HEBE for the STSF hypothesis.

The easiest way to approach the STSF is to consider that the forcing and relaxation times $\tau(\underline{x})$ and sensitivities $\lambda(\underline{x})$ are stochastic fields that are statistically homogeneous in space so that the correlation functions can be written: $R\left(\underline{x}_1, \underline{x}_2, \omega\right) = R\left(\underline{x}, \underline{x} - \underline{\Delta x}, \omega\right) = R\left(\underline{\Delta x}, \omega\right)$. If we assume that the forcing is statistically independent of the temperature, then, taking the high frequency limit of $\hat{G}_T$ in eq. 75:

$$\hat{G}_T\left(\underline{x}, \underline{x} - \underline{\Delta x}, \omega\right) \approx \frac{1}{\left(\tau\left(\underline{x}\right)\tau\left(\underline{x} - \underline{\Delta x}\right)\right)^{1/2} \omega}$$

(86)

we obtain:





$$R_T\left(\underline{\Delta x},\omega\right)=<\left[\frac{\lambda\left(\underline{x}\right)\lambda\left(\underline{x}-\underline{\Delta x}\right)}{\left(\tau\left(\underline{x}\right)\tau\left(\underline{x}-\underline{\Delta x}\right)\right)^{1/2}}\right]>\frac{R_F\left(\underline{\Delta x},\omega\right)}{\omega}$$

(87)

From this, see that if the forcing factorizes $R_F\left(\underline{\Delta x},\omega\right)=R_{F,space}\left(\underline{\Delta x}\right)R_{F,time}\left(\omega\right)$ then the temperature autocorrelation function also factorizes:

$$R_T\left(\underline{\Delta x},\omega\right)=R_{T,space}\left(\underline{\Delta x}\right)R_{T,time}\left(\omega\right);\qquad \begin{aligned}R_{T,space}\left(\underline{\Delta x}\right)&=R_{\lambda\tau^{-1/2}}\left(\underline{\Delta x}\right)R_{F,space}\left(\underline{\Delta x}\right)\\ R_{T,time}\left(\omega\right)&=\frac{R_{F,time}\left(\omega\right)}{\omega}\end{aligned}$$


(88)

Where $R_{\lambda\tau^{-1/2}}\left(\underline{\Delta x}\right)$ is the autocorrelation function of $\lambda\tau^{-1/2}$ (the term in square brackets in eq. 87). From here, the inverse Fourier transform of $R_T\left(\underline{\Delta x},\omega\right)$ and $R_{T,time}\left(\omega\right)$ gives the real space version of the STSF symmetry. Notice that at the STSF hinges on the factorization approximation for $\hat{G}_T\left(\underline{x},\underline{x}-\underline{\Delta x},\omega\right)$ and at low ω, it breaks down.

### Appendix D: Fractional Integration on the sphere

At long enough time scales, the spatial transport of heat is important and the spherical geometry of the Earth must be taken into account. The standard way (see e.g. the review [*North et al.*, 1981]) is to use spherical harmonics. In Appendix 5D of [*Lovejoy and Schertzer*, 2013] these were used to define fractional integrals on the sphere, necessary in order to produce the corresponding multifractal cloud and topography models (see also [*Landais et al.*, 2019]). Spherical harmonics are particularly convenient when the heat transport is diffusive, involving fractional Laplacians. In section 3.5.2, these were

defined in real space by taking the domain of integration to be a sphere. In this appendix we discuss an alternative method of spherical fractional integration that may have theoretical and practical advantages.

The Laplacian on a sphere ($\nabla_\Omega^2$) is the angular part of the Laplacian in spherical coordinates, it is obtained by expressing the Laplacian in spherical coordinates and setting the radial derivatives to zero:

$$\nabla_\Omega^2=\left[\frac{\partial}{\partial\mu}\left(1-\mu^2\right)\frac{\partial}{\partial\mu}+\frac{1}{\left(1-\mu^2\right)}\frac{\partial^2}{\partial\phi^2}\right];\quad \mu=\cos\theta$$

(89)





where θ is the colatitude and $\phi$ is the longitude. The normalized eigenfunctions of $\nabla_\Omega^2$ are the spherical harmonics $Y_{n,m}$:

$$Y_{n,m}(\mu,\phi) = \left[\frac{2n+1}{4\pi}\frac{(n-|m|)!}{(n+|m|)!}\right]^{1/2} P_{n,|m|}(\mu)e^{im\phi}\left(\begin{array}{cc}(-1)^m; & m\geq 0 \\ \\ 1; & m<0\end{array}\right); \quad \mu=\cos\theta; \quad -n\leq m\leq n$$

(90)

With $m$, $n$ integer, $n\geq 0$ and $P_{n,m}$ is the associated Legendre polynomial. $Y_{n,m}$ satisfies:

$$-\nabla_\Omega^2 Y_{n,m}(\mu,\phi) = n(n+1)Y_{n,m}(\mu,\phi)$$

(91)

So that $n(n+1)$ are the eigenvalues. Since $|m|\leq n$ there are $2n+1$ degenerate eigenvalues and functions for each $n$.

The spherical harmonics form a complete orthogonal basis, so that any function $f(\mu,\phi)$ on the sphere can be uniquely expressed in terms of a spherical harmonic expansion:

$$f(\mu,\phi) = \sum_{n=0}^{\infty}\sum_{m=-n}^{n} F_{n,m}^{(0)}Y_{n,m}(\mu,\phi); \quad F_{n,m}^{(0)} = \int_0^{2\pi}\int_{-1}^{1} Y^*_{n,m}(\mu,\phi)f(\mu,\phi)d\mu\,d\phi$$

(92)

Where the $F_{n,m}^{(0)}$ are the coefficients of the expansion without fractional integration (i.e. of order 0, indicated in the superscript). This suggests the following definition for a fractional spherical integration order $H$ of a spherical harmonic:

$$\left(-\nabla_\Omega^2\right)^{-H/2}Y_{n,m}(\mu) = \left[n(n+1)\right]^{-H/2}Y_{n,m}(\mu); \quad n\geq 1$$
,

(93)

for the HEBE, we take $H=1$ which corresponds to the ½ power of the inverse Laplacian. We have excluded the value $n=0$ since when $H>0$, the filter $\left[n(n+1)\right]^{-H/2}$ divergences; since $Y_{0,0}(\mu,\phi) = \frac{1}{\sqrt{4\pi}}$, this component corresponds to the

mean. Therefore the above definition is adequate for mean zero anomalies. Alternatively, the mean can be removed and taken care of separately, see below. With this definition, the fractional integral of the zero mean function $f$ is:

$$\left(-\nabla_\Omega^2\right)^{-H/2}f(\mu,\phi) = \sum_{n=1}^{\infty}\sum_{m=-n}^{n} F_{n,m}^{(H)}Y_{n,m}(\mu,\phi); \quad F_{n,m}^{(H)} = \left[n(n+1)\right]^{-H/2}F_{n,m}^{(0)}$$

(94)





i.e. a filter in spherical harmonic space, analogous to the Fourier filter $\left|k\right|^{-H}$ for an isotropic fractional integration in Cartesian coordinates.

The definition of the fractional Laplacian (eq. 93, 94) is adequate when the horizontal transport coefficients are constant, but in section 3.5, we saw that more generally, the half order divergence operator was written: $l\left(\mu,\phi\right)^{-1}\left(-\nabla_{\Omega}^{2}\right)^{-1/2}$ i.e. there was an extra multiplication by the spatially varying diffusion length $l\left(\mu,\phi\right)$. In flat (Cartesian) coordinates, such real space multiplications correspond to Fourier space convolutions so that this operator can also be conveniently expressed in Fourier space. However, with spherical harmonics, this simplicity is lost: although isotropic real space convolutions can still be performed by filtering the harmonics, real space multiplications no longer correspond to convolutions of harmonic coefficients, the closest spherical harmonic equivalent is much more complicated, it involves Clebsch-Gordon coefficients.

    A method of fractionally integrating the mean ($n = 0$) component was developed for the purpose of multifractal modeling in Appendix 5D of [*Lovejoy and Schertzer*, 2013]. There, a different definition of fractional integrals on the sphere was proposed: a convolution with the function $\Theta^{-(2-H)}$, where $\Theta$ is the angle between two points subtended at the center of the sphere. The function $\Theta^{-(2-H)} / \Gamma(H/2)$ was numerically expanded in spherical harmonics and the convolution was again performed by filtering the coefficients (the constant $\Gamma(H/2)$ is needed so that the normalization is the same as for the definition eq. 92). The main difference between the two definitions is that the latter can be directly applied to fields with nonzero means. With definition, the $H$ order fractional integral of a constant function on the sphere (representing the nonzero mean), is simply the value multiplied by $2^{-H/2}\sqrt{\pi} / \Gamma\left(H/2\right)\int_{0}^{2\pi} s^{-(2-H)}\sin s\,ds$ which for the HEBE $H=1$ case, reduces to $(1/2)^{1/2}Si(2\pi)$ where $Si$ is the standard sine integral function. However for the coefficients $n\geq1$, numerical tests show that the two definitions are almost exactly the same; for example with $H=1$, the spherical harmonic coefficients of $\Theta^{-(2-H)}$ are within 3% for all $n\geq1$ and the ratio converges rapidly to 1 for large $n$. The conclusion is that filtering the anomaly by $\left[n\left(n+1\right)\right]^{-H/2}$ and the multiplying the mean by the above factor is a practical method of fractionally integrating a function on the sphere.

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





**Figures**


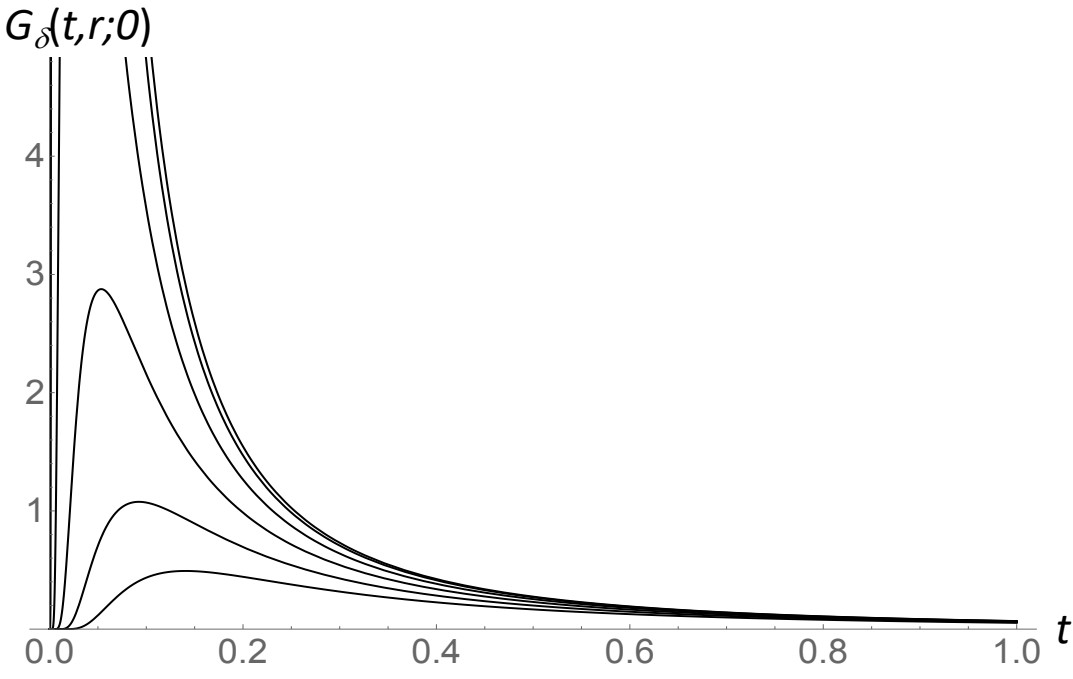

**Fig. 1:** The surface impulse response function ($G_\delta(t,r;0)$, eq. 12, i.e. Dirac in time and Dirac in space) as a function of nondimensional time (*t*) for nondimensional distance from the source increasing from *r* = 0 (top) to *r* = 1 in steps of 0.2 (top to bottom).



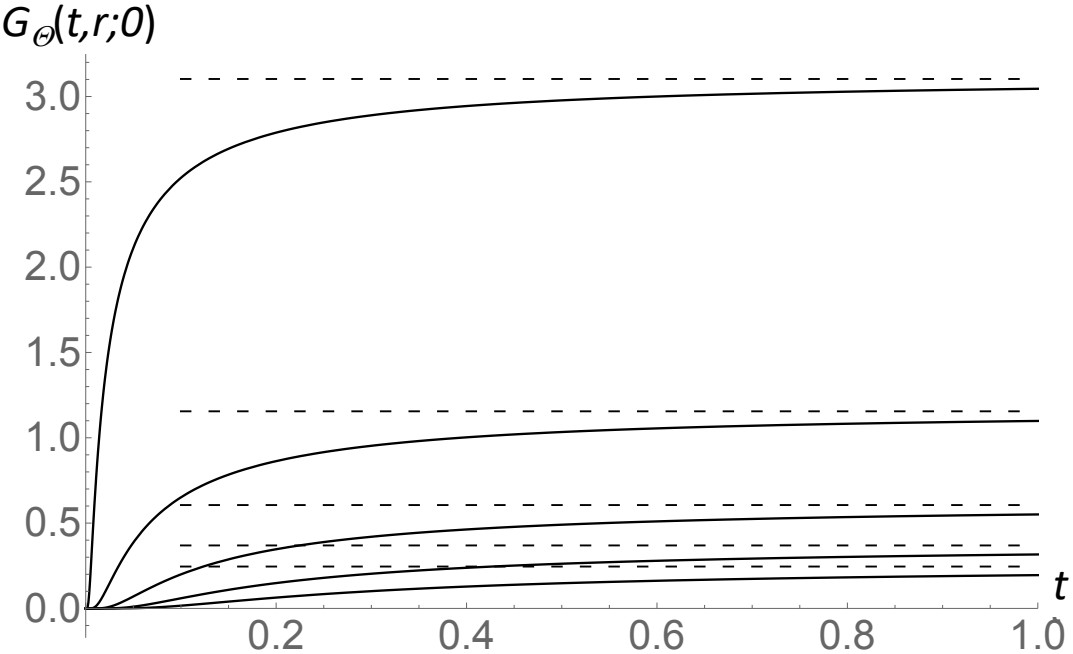


**Fig. 2: The surface step response (time), Dirac (space) function ($G_{\Theta}\left(t,r;0\right)$, eq. 12) as a function of nondimensional time, each curve is for a different nondimensional distance from the source increasing from $r = 0.2$ (top) to $r = 1$ in steps of 0.2 (top to bottom). At each distance $r$, the temperature approaches thermodynamic equilibrium (= $G_{therm,\delta}(r)$, eq. 20) at large $t$ (shown by dashed**
**horizontal lines).**





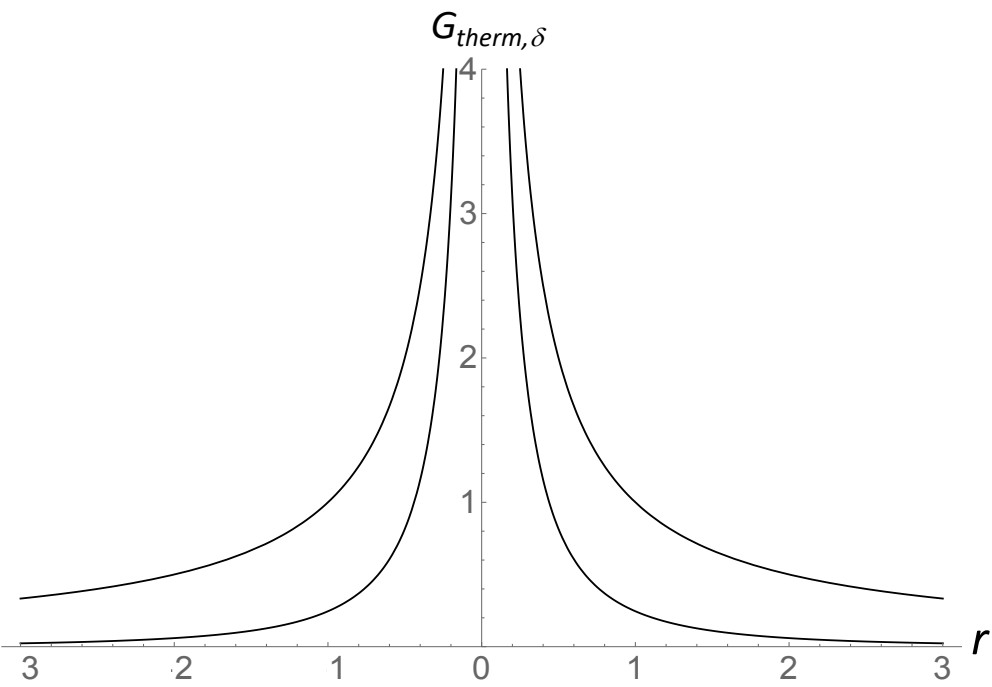

**Fig. 3: A comparison of the spatial impulse response Green's functions for thermal equilibrium with surface forcing via conduction only (i.e.** $\left.\dfrac{\partial T_{therm}}{\partial z}\right|_{z=0} = \delta(\underline{x})$**, no radiation), top** $= r^1$**), and bottom, the same but with conduction – radiative forcing via the surface BC (** $\left.\dfrac{\partial T_{therm}}{\partial z}\right|_{z=0} + T(r;0) = \delta(\underline{x})$**) that is asymptotically** $\approx r^3$ **(eq. 21).**


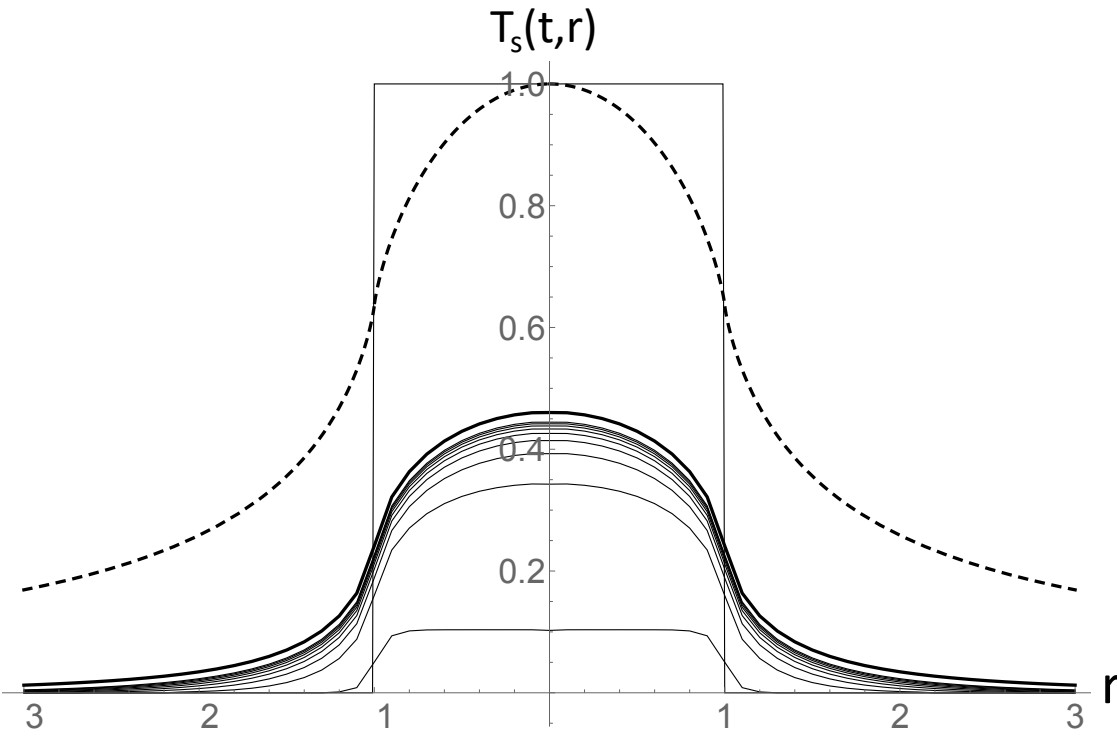

**Fig. 4: This is the step response in time and (circular) step in space for conductive-radiative forcing. Lines for $t = 0.01$ (bottom), 0.2, 0.4, ... 1.6 (black, bottom to top, the thick black line is for $t = \infty$ (thermodynamic equilibrium). The nondimensional forcing is the rectangle (from unit circular forcing). Also shown (top dashed) is the thermodynamic equilibrium when the forcing is purely due to unit conductive heating over the unit circle.**





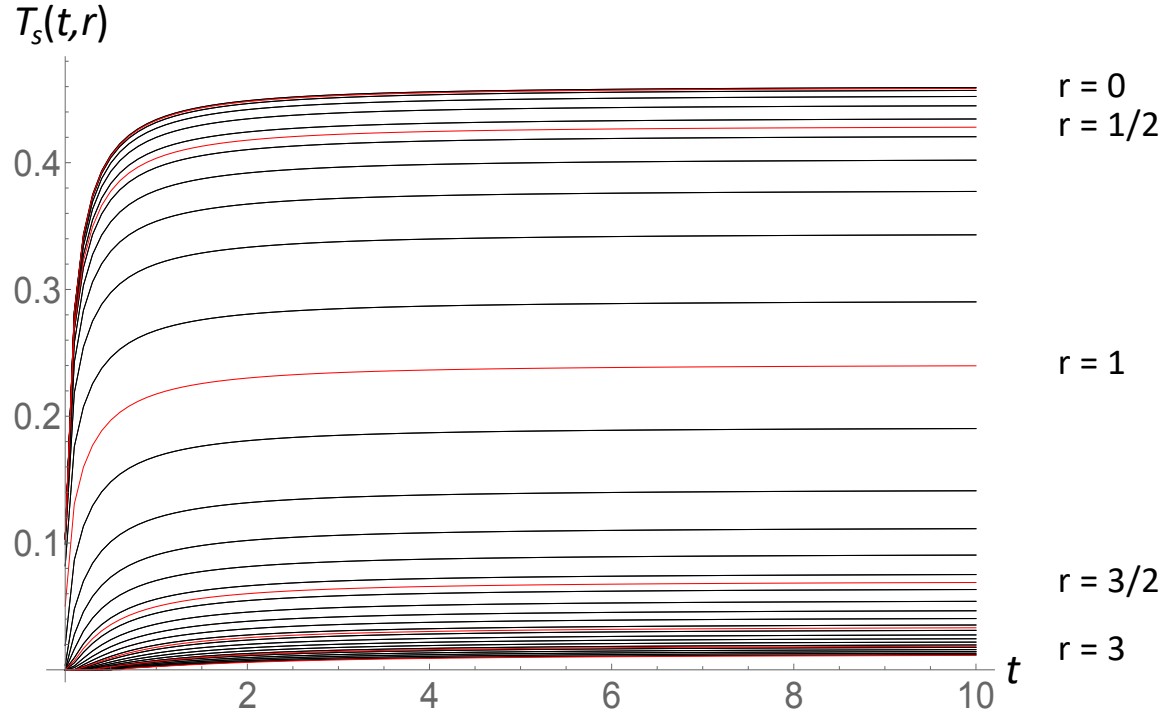


**Fig. 5: The response to a unit intensity forcing in the unit circle. The temperature as a function of nondimensional time is given for different distances from the center top (r = 0) to bottom (r = 3), from the same data as before... red every 1/2, black every 0.1 (top, r = 0, bottom, r = 3).**





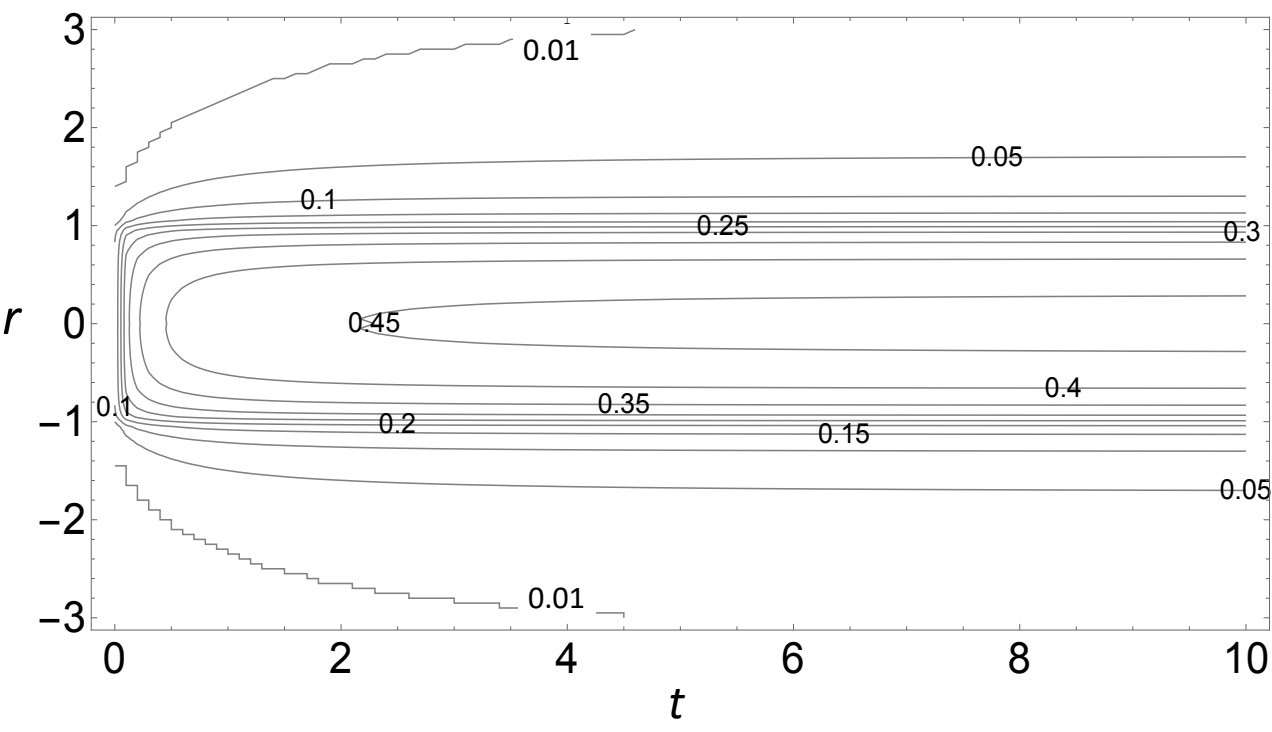


**Fig. 6: Space - time contours for unit circle forcing as a function of nondimensional time (left to right) and nondimensional horizontal distance (vertical axis) and nondimensional time left to right.**


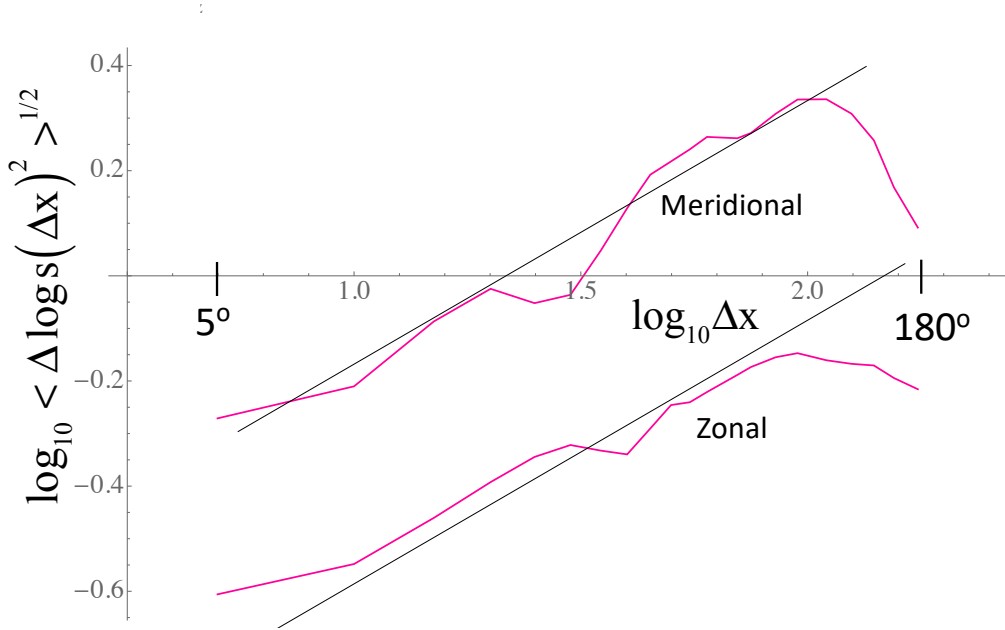


**Fig. 7: The RMS fluctuations (at $\Delta t$ = 1 month resolution)** $\Delta \log s_{\Delta t}(\Delta x)$ **(zonal, bottom),** $\Delta \log s_{\Delta t}(\Delta y)$ **(meridional, top) from NCAR reanalyses. The vertical scale is dimensionless, the horizontal scale is in $\log_{10}$ (degrees) with the minimum (5°) and maximum (180°) indicated in large, bold font. The black lines are reference lines (not regressions) with slopes $H_x = H_y$ =0.5.**



| Parameters | Symbol | Estimated Value |
|---|---|---|
| | | |
| Specific heat per volume | $\rho c$ | $\approx 10^6$ J/m$^3$ |
| Climate sensitivity | $\lambda$ | $\approx 1$ K/(W/m$^2$) |
| Vertical diffusivity (ocean) | $\kappa_v$ | $\approx 10^{-4}$ m$^2$/s |
| Vertical diffusivity (soil) | $\kappa_v$ | $\approx 10^{-6}$ m$^2$/s |
| Horizontal diffusivity | $\kappa_h$ | $\approx 1$ m$^2$/s |
| Vertical Diffusion depth (oceans) | $l_v = \left( \tau \kappa_v \right)^{1/2}$ | $\approx 100$ m |
| Vertical Diffusion depth (soil) | $l_v = \left( \tau \kappa_v \right)^{1/2}$ | $\approx 3 - 10$ m |
| **Relaxation time** | $\tau = \kappa_v \left( \rho c \lambda \right)^2$ | $\approx \mathbf{10^8}$ **s** |
| Horizontal Diffusion length | $l_h = \left( \tau \kappa_h \right)^{1/2}$ | $\approx 10^4$ m |
| **Effective horizontal heat transport velocity** | $V = l_h / \tau$ | $\approx \mathbf{10^{-4}}$ **m/s** |
| Effective advection velocity | $v_h$ | $\approx 10^{-4}$ m/s |
| Nondimension advection velocity | $\alpha$ | 0.1 - 1 |
| Characteristic Zonal variation length | $L_{EW}$ | $\approx 1.5 \times 10^7$ m |
| Characteristic Meridional variation length | $L_{NS}$ | $\approx 3 \times 10^6$ m |

**Table 1: Empirical estimates of the parameters used in this paper; see appendix A for details.**