# Peer review of "The Half-order Energy Balance Equation, Part 2: The inhomogeneous HEBE and 2D energy balance models"

_Earth System Dynamics, 2020_

## Referee Comment (RC1) · Anonymous Referee #1 · 10 Jul 2020

**General comments:** I think this is a notable (two-part) paper. Its key message, that the heat flux at the earth's surface is a derivative of order half of the temperature, and that this modifies the simplest EBMs in an important way is both significant in itself, and provides a foundation for the author's concurrent work on fractional stochastic eneergy balance models.

I have only one gripe that needs attention. It relates to earlier work which needs to be more fully described and integrated into the manuscript. When this is done so it will actually reinforce the author's message, I think.

**Specific comment: Earlier work on half-order derivatives in heat transfer**.

[Figure]

The list of references on fractional calculus seems to me to be comprehensive in general, but to be missing a key reference. Podlubny [1999] notes in his preface that: *... from the viewpoint of applications in physics, chemistry and engineering it was undoubtedly the book written by K. B. Oldham and J. Spanier [i.e. "The Fractional Calculus", Academic Press, 1974; now in a Dover Edition] which played an outstanding role in the development of the subject which can be called applied fractional calculus. Moreover, it was the first book which was entirely devoted to a systematic presentation of the ideas, methods, and applications of the fractional calculus.*

Referring back to this book suggests to me that to say, as the manuscript presently does, that *"... half-order derivatives have occasionally [sic] been used in the context of the heat equation, (at least since [Babenko, 1986])"* substantially underestimates the extent to which half order derivatives have already been studied in the heat equation context. Oldham and Spanier devote their chapter 11 to applications of what they call the semidifferential operator, i.e. the fractional derivative of half order, to diffusion problems including heat transfer.

The book built on their own papers, particularly Oldham KB, Spanier J (1972) A general solution of the diffusion equation for semiinfinite geometries, J Math Anal Appl 39:665–669 and Oldham KB (1973) Diffusive transport to planar, cylindrical and spherical electrodes, J Electroanal Chem Interfacial Electrochem, 41:351–358.

They give the diffusion equation as

$$\frac{\partial}{\partial t} F(\xi, \eta, \zeta, t) = \kappa \nabla^2 F(\xi, \eta, \zeta, t) \qquad (1)$$

and then note that in in three special, semi-infinite, cases this can be simplified so that Laplacian depends only on the radial co-ordinate $r$ and $t$. In the planar case they give:

$$\frac{\partial}{\partial t}F(r,t) - \kappa \frac{\partial^2}{\partial r^2}F(r,t) = 0 \qquad (2)$$

They take the system is initially in equilibrium $F(r,t) = F_0$, for $t < 0, r \geq 0$. An unspecified perturbation occurs at $t = 0$, and for times of interest $t < 0$ it does not affect regions remote from the $r = 0$ boundary. Hence $F(r,t) = F_0$, for $t \leq \tau, r = \infty$, and in the case of planar geometry they derive the solution:

$$\frac{\partial}{\partial r}F(r,t) = -\frac{1}{\sqrt{\kappa}}\frac{\partial^{1/2}}{\partial t^{1/2}}F(r,t) + \frac{F_0}{\sqrt{\pi\kappa t}} \qquad (3)$$

They then go on to consider the problem of 1D heat conduction in a semi-infinite plane, and so look at the heat equation in the form:

$$\frac{\partial}{\partial t}T(r,t) - \frac{K}{\rho\sigma}\frac{\partial^2}{\partial r^2}T(r,t) = 0 \qquad (4)$$

with appropriate boundary conditions of $T(r,0) = 0$ and $T(\infty,t) = 0$.

The heat flux sought is

$$J(t) \equiv -K\frac{\partial}{\partial r}T(0,t) \qquad (5)$$

which they get from their earlier solution for $\partial F(r,t)/\partial r$ by putting $T$ for $F$, $K/\rho\sigma$ for $\kappa$, and using $T_0 = 0$

$$J(T) = -K\frac{\partial}{\partial r}T(0,t) = \sqrt{K\rho\sigma}\frac{\partial^{1/2}}{\partial t^{1/2}}T(0,t) \qquad (6)$$

Because this result, Oldham and Spanier's equation 11.2.10 is closely related to equation 43 in part I of the present ms, I think that it should be explained clearly whether i) the present paper is effectively an illustration of Oldham and Spanier's result in the EBM context, or ii) whether it offers a derivation in a domain to which Oldham and Spanier's result did not apply. Either situation will be important and publishable but readers need to know which applies.

Interestingly, Oldham and Spanier noted that the equation had been obtained by Meyer in 1960 in a Canadian NRC technical report ("A heat-flux-meter for use with thin film surface thermometers"), but rather than being written as a half order derivative it was then given in the alternative integral form

$$J(T) = \sqrt{\frac{K\rho\sigma}{4\pi}} \Big[\frac{2T(0,t)}{\sqrt{t}} + \int_0^t \frac{T(0,t) - T(0,\tau)}{\sqrt{t-\tau}} d\tau\Big] \tag{7}$$

without explicitly using fractional calculus. It was thus known in the heat transfer context even before the first EBMs were derived, in a sense reinforcing the present author's point.

---

## Referee Comment (RC2) · Anonymous Referee #2 · 13 Jul 2020

This second part reviewed here extends the approach of Part 1 to higher spatial dimension and inhomogeneous thermal models of the earth's response to radiative forcing. There is an appropriate summary of Part 1 that puts the new contribution into context. The full model considered here includes varying horizontal and vertical thermal diffusivities, thermal capacities, sensitivities and spatio-temporal forcing. By a heuristic method of Babenko, the author expands the inhomogeneous operator to give 2D energy balance equations that will be useful for studying spatio-temporal responses to forcing. The manuscript includes a number of appendices that examine horizontal structures, cross-correlations, space-time factorization of quantities such as autocorrelation and that extends the results from flat space to the sphere. The analysis seems

to be carefully done, and care is taken to distinguish cases where there may not be a rigorous justification. I would be interested to see a bit more discussion of the "bottom boundary condition" T=0 at z=-infinity. I think it would also be useful to include some discussion of how atmosphere/ocean convection is/is not represented in the model.

---

## Author Comment (AC1) · 17 Jul 2020

Interactive comment on "The Half-order EnergyBalance Equation, Part 2:The inhomogeneousHEBE and 2D energy balance models"

FOR THE EQUATIONS SEE THE SUPPLEMENT

Anonymous Referee #1

General comments: I think this is a notable (two-part) paper. Its key message, that the heat flux at the earth's surface is a derivative of order half of the temperature, and that this modifies the simplest EBMs in an important way is both significant in itself, and

provides a foundation for the author's concurrent work on fractional stochastic energy balance models.

Au: Thank you for the enthusiastic review!

I have only one gripe that needs attention. It relates to earlier work which needs to be more fully described and integrated into the manuscript. When this is done so it will actually reinforce the author's message, I think.

Au: The Oldham references are quite useful, thanks! I respond in more detail below.

Specific comment: Earlier work on half-order derivatives in heat transfer The list of references on fractional calculus seems to me to be comprehensive in general, but to be missing a key reference. Podlubny [1999] notes in his preface that:... from the viewpoint of applications in physics, chemistry and engineering it was undoubtedly the book written by K. B. Oldham and J. Spanier [i.e. "The Fractional Calculus", Academic Press, 1974; now in a Dover Edition] which played an outstanding role in the development of the subject which can be called applied fractional calculus. Moreover, it was the first book which was entirely devoted to a systematic presentation of the ideas, methods, and applications of the fractional calculus.

Referring back to this book suggests to me that to say, as the manuscript presently does, that"... half-order derivatives have occasionally [sic] been used in the context of the heat equation, (at least since [Babenko, 1986]) "substantially underestimates the extent to which half order derivatives have already been studied in the heat equation context. Oldham and Spanier devote their chapter 11 to applications of what they call the semi differential operator, i.e. the fractional derivative of half order, to diffusion problems including heat transfer.

The book built on their own papers, particularly Oldham KB, Spanier J (1972) A general solution of the diffusion equation for semi infinite geometries, J Math Anal Appl 39:665–669 and Oldham KB (1973) Diffusive transport to planar, cylindrical and spher-ical
electrodes, J Electroanal Chem Interfacial Electrochem, 41:351–358. They give the diffusion equation as:

and then note that in three special, semi-infinite, cases this can be simplified so that Laplacian depends only on the radial co-ordinate r and t. In the planar case they give:

They take the system is initially in equilibrium F(r,t) =F0, for t <0,r≥0. An unspecified perturbation occurs at t= 0, and for times of interest t <0 it does not affect regions remote from the r= 0 boundary. Hence F(r,t) =F0, for t≤$\tau$,r=$\infty$, and in the case of planar geometry they derive the solution:

They then go on to consider the problem of 1D heat conduction in a semi-infinite plane, and so look at the heat equation in the form:

with appropriate boundary conditions of T(r,0) = 0 and T($\infty$,t) = 0.The heat flux sought is

which they get from their earlier solution for $\partial$F(r,t)/$\partial$r by putting T for F,K/$\sigma$ for $\kappa$,and using

Because this result, Oldham and Spanier's equation 11.2.10 is closely related to equation 43 in part I of the present ms, I think that it should be explained clearly whether i) the present paper is effectively an illustration of Oldham and Spanier's result in the EBM context, or ii) whether it offers a derivation in a domain to which Oldham and Spanier's result did not apply. Either situation will be important and publishable but readers need to know which applies. Interestingly, Oldham and Spanier noted that the equation had been obtained by Meyerin 1960 in a Canadian NRC technical report ("A heat-flux-meter for use with thin film surface thermometers"), but rather than being written as a half order derivative it was then given in the alternative integral form:

without explicitly using fractional calculus. It was thus known in the heat transfer context even before the first EBMs were derived, in a sense reinforcing the present author's point.

Au: There are several important differences w.r.t. to Oldham's results.

a) Oldham considers only a single spatial degree of freedom r corresponding to either the "zero-dimensional" model (eq. 22 part 1) or cylindrical or spherical geometries that we do not consider. He nowhere considers fractional space-time operators as in part 2. I.e. he neither treats homogeneous operators but with inhomogeneous boundary conditions, nor does Oldham treat inhomogeneous media (inhomogeneous transport operators). In other words essentially all of part 2 (eq. 3 and later) is outside his scope.

b) Our boundary radiative-conductive boundary conditions are special cases of "Robin" boundary conditions i.e. they involve a linear combination of the field and it's normal gradient over a surface. Although Robin boundary conditions are occasionally used in insulating boundary condition problems in convective diffusive equations, they are not identical to the radiative-conductive conditions used here. Oldham mentions Cauchy, Neumann and Dirichlet boundary conditions and says that "any other type" could be used. In other words he realized that his formalism was more general than the applications he developed, but did not pursue these. I will add this information in the revised ms.

c). Although it is not essential, Oldham's application of the method was to use more or less standard boundary conditions (Dirichlet) and then deduce the heat flux across surfaces from this. As far as I can tell, since then, this is almost invariably the way the method has been applied.

d) A final more minor difference is that we also treated the Weyl derivative and used the corresponding Fourier techniques.

We will add references to these difference in the new ms.

Please also note the supplement to this comment:
https://esd.copernicus.org/preprints/esd-2020-13/esd-2020-13-AC1-supplement.pdf

[Figure]

**Supplement:**

**Interactive comment on**
**"The Half-order EnergyBalance Equation, Part 2:The inhomogeneousHEBE and 2D energy balance models"**

**Anonymous Referee #1**

General comments: I think this is a notable (two-part) paper. Its key message, that the heat flux at the earth's surface is a derivative of order half of the temperature, and that this modifies the simplest EBMs in an important way is both significant in itself, and provides a foundation for the author's concurrent work on fractional stochastic energy balance models.

*Au: Thank you for the enthusiastic review!*

I have only one gripe that needs attention. It relates to earlier work which needs to be more fully described and integrated into the manuscript. When this is done so it will actually reinforce the author's message, I think.

*Au: The Oldham references are quite useful, thanks!  I respond in more detail below.*

**Specific comment: Earlier work on half-order derivatives in heat transfer**
The list of references on fractional calculus seems to me to be comprehensive in general, but to be missing a key reference. Podlubny [1999] notes in his preface that:... from the viewpoint of applications in physics, chemistry and engineering it was undoubtedly the book written by K. B. Oldham and J. Spanier [i.e. "The Fractional Calculus", Academic Press, 1974; now in a Dover Edition] which played an outstanding role in the development of the subject which can be called applied fractional calculus.  Moreover, it was the first book which was entirely devoted to a systematic presentation of the ideas, methods, and applications of the fractional calculus.

Referring back to this book suggests to me that to say, as the manuscript presently does, that"... half-order derivatives have occasionally [sic] been used in the context of the heat equation, (at least since [Babenko, 1986]) "substantially underestimates the extent to which half order derivatives have

already been studied in the heat equation context. Oldham and Spanier devote their chapter 11 to applications of what they call the semi differential operator, i.e. the fractional derivative of half order, to diffusion problems including heat transfer.

The book built on their own papers, particularly *Oldham KB, Spanier J (1972) A general solution of the diffusion equation for semi infinite geometries, J Math Anal Appl 39:665–669* and *Oldham KB (1973) Diffusive transport to planar, cylindrical and spher-ical electrodes, J Electroanal Chem Interfacial Electrochem, 41:351–358*. They give the diffusion equation as:

$$\frac{\partial}{\partial t} F(\xi, \eta, \zeta, t) = \kappa \nabla^2 F(\xi, \eta, \zeta, t) \tag{1}$$

and then note that in three special, semi-infinite, cases this can be simplified so that Laplacian depends only on the radial co-ordinate r and t. In the planar case they give:

$$\frac{\partial}{\partial t} F(r, t) - \kappa \frac{\partial^2}{\partial r^2} F(r, t) = 0 \tag{2}$$

They take the system is initially in equilibrium $F(r,t) = F_0$, for t <0, r≥0. An unspecified perturbation occurs at t= 0, and for times of interest t <0 it does not affect regions remote from the r= 0 boundary. Hence $F(r,t) = F_0$, for t≤τ, r=∞, and in the case of planar geometry they derive the solution:

$$\frac{\partial}{\partial r} F(r, t) = -\frac{1}{\sqrt{\kappa}} \frac{\partial^{1/2}}{\partial t^{1/2}} F(r, t) + \frac{F_0}{\sqrt{\pi \kappa t}} \tag{3}$$

They then go on to consider the problem of 1D heat conduction in a semi-infinite plane, and so look at the heat equation in the form:

$$\frac{\partial}{\partial t} T(r, t) - \frac{K}{\rho \sigma} \frac{\partial^2}{\partial r^2} T(r, t) = 0 \tag{4}$$

with appropriate boundary conditions of T(r,0) = 0 and T(∞,t) = 0. The heat flux sought is

$$J(t) \equiv -K \frac{\partial}{\partial r} T(0, t) \tag{5}$$

which they get from their earlier solution for $\partial F(r,t)/\partial r$ by putting T for F, $K/\rho\sigma$ for $\kappa$, and using

$$J(T) = -K \frac{\partial}{\partial r} T(0, t) = \sqrt{K\rho\sigma} \frac{\partial^{1/2}}{\partial t^{1/2}} T(0, t) \tag{6}$$

Because this result, Oldham and Spanier's equation 11.2.10 is closely related to equation 43 in part I of the present ms, I think that it should be explained clearly whether i) the present paper is effectively an illustration of Oldham and Spanier's result in the EBM context, or ii) whether it offers a derivation in a domain to which Oldham and Spanier's result did not apply. Either situation will be important and publishable but readers need to know which applies. Interestingly, Oldham and Spanier noted that the equation had been obtained by Meyerin 1960 in a Canadian NRC technical report ("A heat-flux-meter for use with thin film surface thermometers"), but rather than being written as a half order derivative it was then given in the alternative integral form:

$$J(T) = \sqrt{\frac{K\rho\sigma}{4\pi}} \left[ \frac{2T(0, t)}{\sqrt{t}} + \int_0^t \frac{T(0, t) - T(0, \tau)}{\sqrt{t - \tau}} d\tau \right] \tag{7}$$

without explicitly using fractional calculus. It was thus known in the heat transfer context even before the first EBMs were derived, in a sense reinforcing the present author's point.

*Au: There are several important differences w.r.t. to Oldham's results.*

*a) Oldham considers only a single spatial degree of freedom r corresponding to either the "zero-dimensional" model (eq. 22 part 1) or cylindrical or spherical geometries that we do not consider. He nowhere considers fractional space-time operators as in part 2. I.e. he neither treats homogeneous operators but with inhomogeneous boundary conditions, nor does Oldham treat inhomogeneous media (inhomogeneous transport operators). In other words essentially all of part 2 (eq. 3 and later) is outside his scope.*

*b)  Our boundary radiative-conductive boundary conditions are special cases of "Robin" boundary conditions i.e. they involve a linear combination of the field and it's normal gradient over a surface.  Although Robin boundary conditions are occasionally used in insulating boundary condition problems in convective diffusive equations, they are not identical to the radiative-conductive conditions used here.   Oldham mentions Cauchy, Neumann and Dirichlet boundary conditions and says that "any other type" could be used.  In other words he realized that his formalism was more general than the applications he developped, but did not pursue these.  I will add this information in the revised ms.*

*c). Although it is not essential,  Oldham's application of the method was to use more or less standard boundary conditions (Dirichlet) and then deduce the heat flux across surfaces from this.  As far as I can tell,  since then, this is almost invariably the way the method has been applied.*

*d) A final more minor difference is that we also treated the Weyl derivative and used the corresponding Fourier techniques.*

*We will add references to these difference in the new ms.*

---

## Author Comment (AC2) · 17 Jul 2020

Interactive comment on "The Half-order EnergyBalance Equation, Part 2:The inhomogeneousHEBE and 2D energy balance models"

Anonymous Referee #2

This second part reviewed here extends the approach of Part 1 to higher spatial dimension and inhomogeneous thermal models of the earth's response to radiative forcing.There is an appropriate summary of Part 1 that puts the new contribution into con-

text. The full model considered here includes varying horizontal and vertical thermal diffusivities, thermal capacities, sensitivities and spatio-temporal forcing. By a heuristic method of Babenko, the author expands the inhomogeneous operator to give 2D energy balance equations that will be useful for studying spatio-temporal responses to forcing. The manuscript includes a number of appendices that examine horizontal structures, cross-correlations, space-time factorization of quantities such as autocorrelation and that extends the results from flat space to the sphere. The analysis seems to be carefully done, and care is taken to distinguish cases where there may not be a rigorous justification.

Au: I thank the referee for the very positive review!

I would be interested to see a bit more discussion of the "bottom boundary condition" T=0 at z=-infinity. I think it would also be useful to include some discussion of how atmosphere/ocean convection is/is not represented in the model.

Au: The role of the bottom boundary condition was addressed in part I where (just after eq. 29) it is shown that the influence of the bottom BC decays exponentially quickly with depth so that below a few diffusion depths it is essentially irrelevant. In oceans this would likely imply depths of hundreds of meters. In part I we will add some new material clarifying the nature of the surface.